# Decentralized and Disentangled Task–Role Representation Learning for Generalizable Offline Multi-Agent Meta Reinforcement Learning

**Lei Yuan** [1 2 3]  **Ruiqi Xue** [1 2]  **Yang Yu** [1 2 3]

## Abstract

Offline meta reinforcement learning (RL) enables agents to learn a unified policy from multi-task offline data to support generalization in out-of-distribution (OOD) tasks. Recent approaches in single-agent RL tackle this by learning an efficient task representation to distinguish between tasks, showing promising adaptation ability. However, when extended to multi-agent settings, these methods struggle with decentralized task identification due to limited global information, and suffer from inefficient knowledge transfer in the absence of role information. To address this, we propose $D^2TR$, a novel context-based meta RL framework with efficient decentralized and disentangled task-role identification. Specifically, $D^2TR$ first introduces mutual information knowledge distillation to align decentralized task representations with centralized task representations inferred from global trajectories, enabling efficient decentralized team-centric information identification. Next, $D^2TR$ leverages a large language model to assign semantic roles to trajectories in offline data, and achieves effective individual-centric information inference by learning decentralized role representations. Extensive experiments conducted on commonly used multi-agent environments, including CN, SMAC, and SMACv2, demonstrate that $D^2TR$ exhibits strong generalization performance to unseen tasks, outperforming prior multi-agent multi-task and context-based meta RL baselines.

## 1. Introduction

Multi-agent reinforcement learning (MARL) (Albrecht et al., 2024) has attracted increasing attention, achieving notable success in cooperative domains such as autonomous driving (Zhang et al., 2024b), financial trading (Sarin et al., 2024), and embodied intelligence (Feng et al., 2025). However, most existing MARL methods are designed for closed-world settings (Oroojlooy & Hajinezhad, 2023), where a fixed set of agents is trained to solve a single task under a specific environment configuration. This assumption limits its real-world applicability, as agents must often collaborate under varying team compositions and across diverse tasks, which necessitates research on open-environment MARL (Yuan et al., 2023). For example, in embodied intelligence, humanoid robots may need to coordinate with fluctuating numbers of teammates to transport objects of varying shapes and sizes to different targets (Liu et al., 2025b).

Meta RL offers a promising pathway toward open-environment RL by enabling policies to rapidly adapt and generalize to unseen tasks in a few-shot manner (Beck et al., 2023). While foundational works have successfully established recurrent (Mishra et al., 2018), gradient-based (Finn et al., 2017), and context-based (Rakelly et al., 2019) mechanisms in online settings, there is a growing focus on training agents utilizing solely pre-collected datasets. This offline setting differs fundamentally from the online paradigm by precluding environmental interaction for system identification, thereby introducing severe challenges such as out-of-distribution actions and context shifts, particularly where static data fails to cover optimal behaviors (Dorfman et al., 2021). To address these complexities, context-based offline meta RL has gained prominence; by training a context encoder to extract task-relevant information, this approach achieves high sample efficiency and strong empirical performance (Li et al., 2021b; Gao et al., 2023; Li et al., 2024).

However, when extended to multi-agent settings, these methods encounter two new challenges. First, the decentralized execution paradigm inherent to multi-agent systems requires the context encoder to perform efficient and accurate task identification based solely on decentralized trajectories. Such trajectories often lack critical global information, which can significantly complicate representation learning

[1] National Key Laboratory for Novel Software Technology, Nanjing University [2] School of Artificial Intelligence, Nanjing University [3] Polixir Technologies.. Correspondence to: Yang Yu <yuy@nju.edu.cn>.

*Proceedings of the $43^{rd}$ International Conference on Machine Learning*, Seoul, South Korea. PMLR 306, 2026. Copyright 2026 by the author(s).

and lead to biased or unreliable task inference. Second, the emergence of distinct roles among agents—each specializing in different subtasks—is a fundamental characteristic of multi-agent systems (Becht et al., 1999; Pavón & Gómez-Sanz, 2003; Lhaksmana et al., 2018), roles that are shared across tasks constitute an essential source of transferable knowledge in multi-task learning. However, relying exclusively on team-specific task encodings makes the learning process highly susceptible to role collapse, where agents either converge to homogeneous behaviors or are restricted to rigid cooperation patterns, ultimately degrading generalization performance. Moreover, role information is not explicitly observable in the data; instead, it is embedded in high-level semantic patterns of agent behavior. Effectively extracting and disentangling such latent role representations therefore remains a critical and nontrivial challenge.

To address these challenges, we introduce **D**ecentralized and **D**isentangled **T**ask–**R**ole Representation Learning (D$^2$TR), a novel context-based offline multi-agent meta RL framework that enhances adaptivity and generalizability through improved representation learning. On one hand, we first propose task representation learning with mutual information knowledge distillation, which first trains a centralized task representation with global information, and then aligns the decentralized task representation with the centralized one with mutual information maximization, enabling efficient task-specific encoding across agents. Next, we leverage a large language model (LLM) to extract the underlying behavioral logic of the data at a higher semantic level, thereby assigning roles to trajectories in the offline data, allowing us to perform contrastive learning with these role labels to learn a role encoder to capture individual-centric information from decentralized contexts. Finally, both the task and role encodings will be fed into a Transformer-based policy to perform task-role aware offline multi-agent policy learning. Extensive experiments conducted on standard multi-agent environments, including CN, SMAC, and SMACv2, demonstrate that D$^2$TR achieves superior generalization performance on unseen tasks, outperforming prior multi-agent multi-task and context-based meta-learning baselines.

## 2. Related Work

**Meta RL**  Meta RL trains policies across related tasks to extract shared knowledge, enhancing adaptation and generalization to unseen tasks. Context-based meta RL, known for its sample efficiency and empirical performance, has gained popularity. PEARL (Rakelly et al., 2019) is one of the first to introduce a probabilistic context encoder for task identification in online RL. To address the high cost of trial-and-error in online exploration, attention has shifted to context-based offline meta RL. Approaches like FOCAL (Li et al., 2021b) and CORRO (Yuan & Lu, 2022) improve task

representation by learning encoders using contrastive losses, boosting discriminability and robustness. CSRO (Gao et al., 2023) separates the encoder from the offline behavior policy, while UNICORN (Li et al., 2024) unifies these methods into a mutual information-based framework, adding a decoder loss for theoretical completeness. However, these methods are designed for single-agent settings and struggle in multi-agent scenarios due to limited access to global information. MG2L (Zhao et al., 2025) addresses this by introducing a global task encoder in online multi-agent meta RL. By concatenating global and local encodings during contrastive learning, MG2L reduces the challenges of representation search. While MG2L excels in online settings, its performance drops in offline scenarios with limited context data. The global encoding, while easing contrastive learning, can overshadow local information, limiting decentralized task identification and reducing effectiveness in cooperation.

**Offline and Multi-task MARL**  Following the centralized training with decentralized execution (CTDE) paradigm (Amato, 2024), MARL has made significant strides in solving cooperative problems through decentralized actor learning (Yu et al., 2022) and value decomposition (Sunehag et al., 2018; Rashid et al., 2020). To address the high cost of online trial-and-error, several approaches focus on offline learning. For instance, CFCQL (Shao et al., 2023) uses the conservative operator from CQL (Kumar et al., 2020) to mitigate value overestimation in decomposed Q-networks, while OMIGA (Wang et al., 2023) extends IQL (Kostrikov et al., 2022) with expectile regression to learn policy-independent Q-functions, extracting policies via advantage-weighted behavior cloning. However, these methods are limited to fixed tasks with predefined team configurations. Further work, such as UPDeT (Hu et al., 2021), ODIS (Zhang et al., 2022), HiSSD (Liu et al., 2025a), and BiKT (Zhang et al., 2026), introduces Transformer-based architectures and transferable skills to handle varying team sizes and tasks. Despite their advances, they still struggle with tasks with different cooperative objectives. More related work can be seen in Appendix A.

## 3. Preliminaries

We consider the problem of offline multi-agent meta RL in cooperative multi-agent games under Decentralized Partially Observable Markov Decision Process (Dec-POMDP) (Oliehoek et al., 2016). A Dec-POMDP can be defined as a tuple $\mathcal{M} = \langle \mathcal{N}, \mathcal{S}, \mathcal{O}, \mathcal{A}, \Omega, P, R, \gamma \rangle$, where $\mathcal{N} = \{1, 2, \ldots, n\}$ is the set of agents, $\mathcal{S}$ is the global state space, $\mathcal{O} = \prod_{i=1}^{n} O^i$ denotes the joint observation space, $\mathcal{A} = \prod_{i=1}^{n} A^i$ is the joint action space. $P : \mathcal{S} \times \mathcal{A} \rightarrow \Pr(\mathcal{S})$ is the transition function, $\Omega : \mathcal{S} \times \mathcal{N} \rightarrow \mathcal{O}$ is the observation function, $R : \mathcal{S} \times \mathcal{A} \rightarrow \mathbb{R}$ is the reward function, and $\gamma \in [0, 1)$ is the discount factor. At each time step $t$,

agent $i \in \mathcal{N}$ receives an observation $o_t^i = \Omega(s_t, i) \in O^i$ and outputs an action $a_t^i \in A^i$ with policy $\pi^i(\cdot | o^i)$. The joint action $\boldsymbol{a}_t = (a_t^1, ..., a_t^n)$ leads to the next state $s_{t+1} \sim P(\cdot | s_t, \boldsymbol{a}_t)$ and a shared team reward $r_t = R(s_t, \boldsymbol{a}_t)$. The objective of the policy learning is to find a joint policy $\boldsymbol{\pi}(\cdot | \boldsymbol{o}) = \prod_i \pi^i(\cdot | o^i)$ that maximizes the expected return $\mathbb{E}_{\boldsymbol{\pi}} \left[ \sum_t \gamma^t R(s_t, \boldsymbol{a}_t) \right]$.

Similar to previous offline meta RL formulations, offline multi-agent meta RL involves a task distribution $p(\mathcal{M})$, where each task $M \in p(\mathcal{M})$ is a Dec-POMDP. For different tasks, both the joint observation space $\mathcal{O}_M$, action space $\mathcal{A}_M$, transition function $P_M$, and reward function $R_M$ may vary. During training, agents cannot interact with the environment directly. Instead, they rely on a static dataset $\mathcal{D}$, containing data from $T$ training tasks $\mathcal{M}_{\text{train}} = \{M_1, \ldots, M_T\}$ sampled from $p(\mathcal{M})$. Thus, the training objective would be

$$\max_{\boldsymbol{\pi}} \mathbb{E}_{\pi, M \sim \mathcal{T}_{\text{train}}} \left[ \sum_t \gamma^t R_M(s_t, \boldsymbol{a}_t) \right]. \qquad (1)$$

During deployment, the meta-trained policy encounters a test task $M_{\text{test}} \sim p(\mathcal{M})$. While the policy does not have direct access to the explicit task identity of $M_{\text{test}}$, it can either be provided with decentralized trajectories from $M_{\text{test}}$ for each agent as contexts, or interact with the corresponding online environment for a limited number of steps to actively collect trajectories as contexts. The policy must effectively leverage these contexts to achieve efficient task identification and generalization.

## 4. Method

This section gives the detailed D$^2$TR, a novel algorithm for generalizable offline multi-agent meta RL (Figure 1). Section 4.1 presents D$^2$TR's procedure for task encoder learning, Section 4.2 illustrates the process of role labeling and role encoder learning, while Section 4.3 introduces D$^2$TR's overall policy learning algorithm.

### 4.1. Decentralized Task Representation Learning with Mutual Information Knowledge Distillation

In multi-agent meta RL, task variations arise from differences in cooperative goals or transition dynamics. When cooperative objectives change, optimal action distributions for the same observations can differ significantly, making task representations crucial for conveying intent and enabling coordination. However, the decentralized nature of multi-agent systems requires task representations to be inferred independently, based solely on each agent's local context. To address this challenge, we propose an offline decentralized approach to task representation learning.

Specifically, given an offline dataset $\mathcal{D}_M =$

$\{\tau_1^1, \tau_1^2, \ldots, \tau_1^n, \tau_2^1, \ldots \tau_2^n, \ldots \tau_m^n\}$ for task $M$, where $\tau_i^j$ denotes the individual trajectory of agent $j$ in the $i$-th joint trajectory. We first introduce an individual task encoder $f_I$, which is used to obtain the decentralized task encoding $z_i^j = f_I(\tau_i^j)$ for the given task. To ensure that it captures sufficient task-relevant cooperative objective information, we optimize it by maximizing the mutual information between the encoding and the task

$$\max_{f_I} I(z_i^j; M). \qquad (2)$$

However, due to the absence of global information in individual contexts, directly optimizing the above objective is challenging. To overcome this, we draw inspiration from a classic POMDP-solving approach—knowledge distillation (Gou et al., 2021). Specifically, we first introduce a global task encoder $f_G$ to obtain a global task encoding for the task

$$z_i = f_G(\tau_i^1, \ldots, \tau_i^n), \qquad (3)$$

and maximize the mutual information between the global encoding and task $M$

$$\max_{f_G} I(z_i; M). \qquad (4)$$

Based on this global encoding, we can modify the optimization objective of the individual task encoder to

$$\max_{f_I} I(z_i^j; M) + \alpha I(z_i^j; z_M), \qquad (5)$$

where $z_M$ denotes the global encoding of trajectories sampled from task $M$, and $\alpha$ is a hyperparameter that controls the extent of guidance provided by knowledge distillation. For the above objective, we have the following theorem:

**Theorem 4.1.** *The mutual information knowledge distillation term $I(z_i^j; z_M)$ is a gap-controlled surrogate of the target objective $I(z_i^j; M)$ and the upper-gap term is reduced by maximizing $I(z_M; M)$.*

Therefore, in order to learn the task encoder, our primary goal is to maximize the three mutual information: $I(z_i; M)$, $I(z_i^j; M)$, and $I(z_i^j; z_M)$. Inspired by the theoretical analysis of previous works, let $X$ be the context input received by the encoder. We decompose $X$ into two parts: $(X_t, X_b)$, where $X_b$ represents the behavior-related $(o, a)$ part, and $X_t$ represents the task-related $(o', r)$. Let $Z$ be the local or global encoding, so we have:

**Proposition 4.2** (Central Theorem (Li et al., 2024)). *Ignoring the constants that are unrelated to $Z$, we have*

$$I(Z; X_t | X_b) \leq I(Z; M) \leq I(Z; X_t | X_b) + I(Z; X_b) = I(Z; X).$$
$$(6)$$

Based on the Central Theorem, we obtain

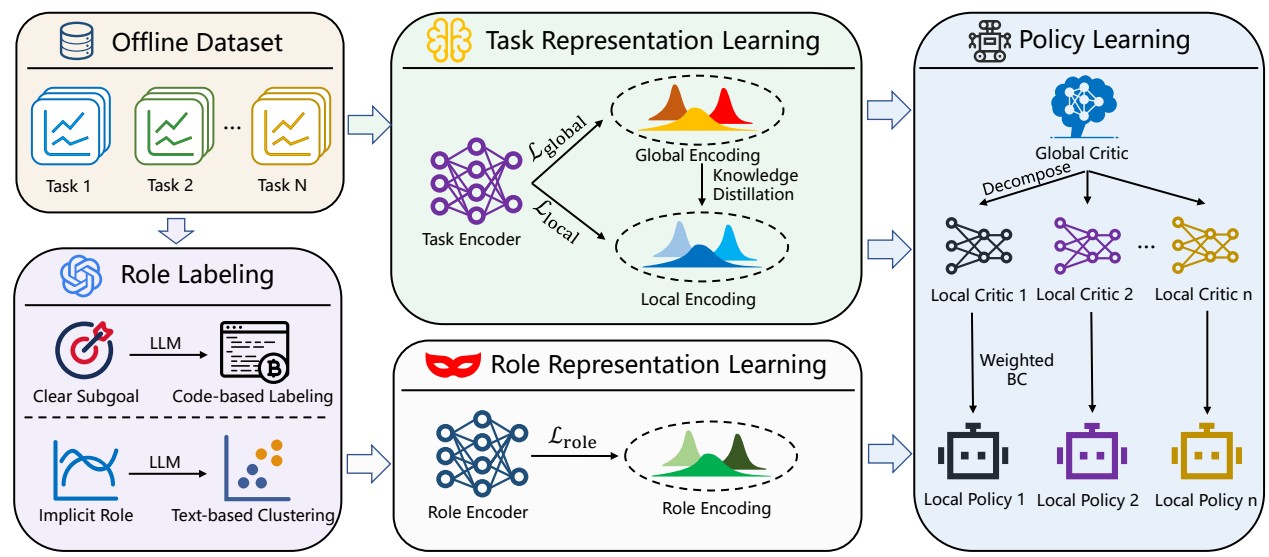

*Figure 1.* The overall workflow of D²TR.

**Theorem 4.3.** *Let $x, x'$ denote the neural network input context sampled in $M$, $x_t$ is $x$'s task-related part while $x_b$ is its behavior related part, $z \sim f(x)$, $f$ is the encoder (can be either $f_I$ or $f_G$), $p_\theta$ is a decoder, and $S$ be a score function. Suppose that $\mathcal{L}_{InfoNCE}(f, M) = -\mathbb{E}_{(x,x') \sim \mathcal{D}_M} \left[ \log \left( \frac{\exp(S(z,z'))}{\sum_{M_i \in \mathcal{M}_{train}} \exp(S(z,z_i))} \right) \right]$, and $L_{recon}(f, M) = -\mathbb{E}_{(x_t, x_b) \sim \mathcal{D}_M, z \sim f(\cdot | x_t, x_b)} [\log p_\theta(x_t | z, x_b)]$, then there exists $\beta \in [0, 1]$ such that*

$$I(Z; M) \approx -\beta L_{InfoNCE}(f, M) - (1-\beta) L_{recon}(f, M). \quad (7)$$

Finally, according to the theorem above, we derive the global task encoder's learning loss as

$$\mathcal{L}_{\text{global}} = \sum_{M \in \mathcal{M}_{\text{train}}} \left( \beta L_{\text{InfoNCE}}(f_G, M) + (1 - \beta) L_{\text{recon}}(f_G, M) \right), \quad (8)$$

and the individual task encoder's learning loss as

$$\mathcal{L}_{\text{local}} = \sum_{M \in \mathcal{M}_{\text{train}}} \left( \beta L_{\text{InfoNCE}}(f_I, M) + (1 - \beta) L_{\text{recon}}(f_I, M) \right.$$
$$\left. - \alpha \mathbb{E}_{(x_I, x_G) \sim \mathcal{D}_M} \left[ \log \left( \frac{\exp(S(f_I(x_I), \text{sg}(f_G(x_G))))}{\sum_{M_i \in \mathcal{M}_{\text{train}}} \exp(S(f_I(x_I), \text{sg}(f_G(x_{G,i}))))} \right) \right] \right), \quad (9)$$

where $x_I$ denotes decentralized contexts, $x_G$ denotes centralized contexts, $x_{G,i}$ denotes centralized contexts sampled in task $M_i$, and $\text{sg}(\cdot)$ denotes stop gradient. The total task encoder learning loss is

$$\mathcal{L}_{\text{task}} = \mathcal{L}_{\text{global}} + \mathcal{L}_{\text{local}}. \quad (10)$$

### 4.2. LLM-assisted Role Representation Learning

With the learned task encoding, the policy gains the ability to recognize team-centric collaboration goals. However,

when only task encoding is available, the policy often resorts to a fixed collaboration mode when faced with unseen tasks, lacking the ability to transfer and reorganize individual behavioral skills. To address this, we propose a paradigm for learning role representation.

**LLM Assisted Role Labeling** To obtain role representations, it is essential that each data point has a corresponding role label. However, role labels are often implicit in the behavioral logic of policies and are challenging to extract directly from raw data. To address this, we propose two role-labeling methods that utilize the high-level semantic reasoning capabilities of LLMs (Lee et al., 2025).

For tasks with explicit subgoals (where each agent's goal is clearly defined), the most efficient and cost-effective approach is to directly provide the task-related information $L_{\text{Task}}$ and the definitions or descriptions of each subgoal $L_{\text{SubGoal}}$ to the LLM, and have it generate a code $c(\cdot)$ to determine which subgoal is most likely associated with a given trajectory.

$$c(\cdot) = \text{LLM}(L_{\text{Task}}, L_{\text{SubGoal}}). \quad (11)$$

With the generated $c(\cdot)$, given behavior-related decentralized trajectories $\tau_{b,i}^j$, where each step contains only behavior-related context $(o_t, a_t)$, the role label can be obtained by

$$R = c(\tau_{b,i}^j). \quad (12)$$

However, for tasks without explicit subgoal definitions, it is challenging for the LLM to directly select the most appropriate subgoal using the code. To address this, we have designed an alternative method for such scenarios. Specifically, for a task $M_i \in \mathcal{M}_{\text{train}}$, we first apply any unsupervised clustering algorithm, such as K-Means, to pre-cluster

the decentralized trajectories in its offline data $\mathcal{D}_{M_i}$, resulting in the pre-clustered decentralized trajectories $P\mathcal{D}_{M_i} = \{\mathcal{D}_{M_i,1}, \ldots, \mathcal{D}_{M_i,K}\}$. Next, we merge the clustering results from all tasks to obtain $P\mathcal{D} = \cup_{M_i} P\mathcal{D}_{M_i}$, which contains a total of $K \times T$ clusters. After that, for each cluster in $P\mathcal{D}$, we sample $E$ trajectories $\{\tau_{M_i,j,1}, \ldots, \tau_{M_i,j,E}\}$ and input them into the LLM to generate descriptions of the behaviors associated with the trajectories

$$L_{M_i,j,e} = \text{LLM}(L_{\text{desc}}, \tau_{M_i,j,e}), e = 1, \ldots, E, \quad (13)$$

where $L_{\text{desc}}$ is the description prompt. Then, all $L_{M_i,j,e}$ will be put into the LLM again to obtain a summary for each cluster

$$L_{M_i,j} = \text{LLM}(L_{\text{sum}}, \{L_{M_i,j,1}, \ldots, L_{M_i,j,E}\}), \quad (14)$$

where $L_{\text{sum}}$ is the summary prompt. Finally, all the prior clusters will be merged based on their summaries

$$C_1, \ldots, C_r = \text{LLM}(L_{\text{merge}}, P\mathcal{D}, \{L_{M_i,j}\}), \quad (15)$$

where $L_{\text{merge}}$ is the merge prompt. All decentralized trajectories in $C_i$ will be assigned with the role label $i$.

**Role Representation Learning**   Supposing the LLM labeled role set is $\mathcal{R} = \{R_1, \ldots, R_r\}$, we can similarly employ mutual information maximization to learn role representations. Specifically, we introduce a role encoder $f_R$, and optimize it similarly with an InfoNCE loss

$$\mathcal{L}_{\text{role}} = -\sum_{R \in \mathcal{R}} \mathbb{E}_{(x_b, x'_b) \sim \mathcal{D}_R} \left[ \log \left( \frac{\exp(S(z_r, z'_r))}{\sum_{R_i \in \mathcal{R}} \exp(S(z_r, z_{r,i}))} \right) \right], \quad (16)$$

where $x_b, x'_b$ are decentralized behavior-related only contexts, $z_r = f_R(x_b), z'_r = f_R(x'_b)$, and $z_{r,i} = f_R(x_{b,i})$ where $x_{b,i}$ is a decentralized behavior-related only context sampled in dataset with respect to role $R_i$.

### 4.3. Task-Role Aware Offline Policy Learning

With the previously learned task and role encodings, our goal is to train a policy that effectively utilizes these encodings for efficient task adaptation and generalization. Previous offline multi-task MARL methods typically use value-based approaches as the backbone, addressing extrapolation error through conservative value estimation. However, we argue that conservative estimation can make the value function overly pessimistic for out-of-distribution (OOD) observations. As a result, using such a value function for decision-making via greedy action selection on unseen tasks may hinder generalization. To overcome this, we propose combining the learned encodings with a policy-based approach to achieve better generalization.

Specifically, we use OMIGA (Wang et al., 2023) to perform weighted behavior cloning, thus constraining the learned

policy to the behavior policy while achieving reward maximization. The optimization objective under the actor-critic framework becomes:

$$\min_{V_i} \mathbb{E}_{\tau^i, \tau, z, z^i, z_r^i} \left[ \exp(\frac{\omega_i(s, z)}{\epsilon} (Q_i(o_i, a_i, z^i, z_r^i) - V_i(o_i, z^i, z_r^i))) + \frac{\omega_i(s, z)V_i(o_i, z^i, z_r^i)}{\epsilon} \right], \quad (17)$$

$$\min_{\substack{Q_i, \omega_i, b \\ i=1,\ldots,n}} \mathbb{E}_{M, \tau^i, \tau, z, z^i, z_r^i} \left[ R_M(s, \boldsymbol{a}) + \gamma V_{tot}(s', \boldsymbol{o}', z, \boldsymbol{z}, \boldsymbol{z_r}) - Q_{tot}(s, \boldsymbol{o}, \boldsymbol{a}, z, \boldsymbol{z}, \boldsymbol{z_r}) \right], \quad (18)$$

$$\max_{\pi_i} \mathbb{E}_{\tau^i, \tau, z, z^i, z_r^i} \left[ \exp(\frac{\omega_i(s, z)}{\epsilon} (Q_i(o_i, a_i, z^i, z_r^i) - V_i(o_i, z^i, z_r^i))) \cdot \log \pi_i(a_i | o_i, z^i, z_r^i) \right], \quad (19)$$

where $\tau^i$ is a decentralized trajectory, $\tau$ is $\tau^i$'s corresponding global trajectory, $z = f_G(\tau)$, $z^i = f_I(\tau^i)$, $z_r^i = f_R(\tau^i)$, $\boldsymbol{z} = [z^1, \ldots, z^n]$, $\boldsymbol{z_r} = [z_r^1, \ldots, z_r^n]$, $\epsilon$ is a hyperparameter. $Q_i, V_i$ are local value functions, $\omega_i, b$ are mixing network parameters, and $Q_{tot}, V_{tot}$ are global value functions, satisfying

$$Q_{tot}(s, \boldsymbol{o}, \boldsymbol{a}, z, \boldsymbol{z}, \boldsymbol{z_r}) = \sum_i \omega_i(s, z)Q_i(o_i, a_i, z^i, z_r^i) + b(s, z),$$

$$V_{tot}(s, \boldsymbol{o}, z, \boldsymbol{z}, \boldsymbol{z_r}) = \sum_i \omega_i(s, z)V_i(o_i, z^i, z_r^i) + b(s, z),$$

$$\omega_i \geq 0, \forall i = 1, \ldots, n. \quad (20)$$

To ensure adaptability to teams with varying numbers of agents, we build all encoders, as well as the policy and value networks, using Transformer or attention-based architectures. Proofs can be seen in Appendix B and more details are provided in Appendix C.

## 5. Experiments

In this section, we conduct a series of experiments to address the following questions: (1) Can D²TR accurately identify tasks, generalize effectively, and outperform the remaining baselines (Section 5.2)? (2) How do the design choices of D²TR contribute to its performance (Section 5.3)? (3) How does the choice and usage of LLMs affect the performance of D²TR (Section 5.4)?

### 5.1. Baselines and Environments

To completely evaluate the performance of D²TR, we introduce the following baselines. First, to demonstrate that our policy backbone (i.e., the variant without encodings) exhibits strong offline multi-task scaling ability, and to highlight the necessity of encodings when cooperation objectives differ across tasks (i.e., rewards conflict between tasks), we

*Table 1.* Overall performance (mean ± std) averaged over 3 seeds. 'IID' and 'OOD' represent win rate in seen and unseen tasks respecitvely. The best result in each environment is highlighted in **bold**.

| | Task | Ours | Backbone | HiSSD | ODIS | UPDeT | FOCAL | CSRO | UNICORN | MG2L |
|---|---|---|---|---|---|---|---|---|---|---|
| | CN Target | **0.74±0.01** | 0.28±0.04 | 0.05±0.01 | 0.02±0.00 | 0.03±0.02 | 0.52±0.00 | 0.17±0.03 | 0.53±0.09 | 0.26±0.04 |
| | CN Agent | 0.89±0.04 | 0.89±0.00 | 0.74±0.05 | 0.59±0.13 | 0.69±0.06 | 0.88±0.01 | 0.86±0.07 | 0.90±0.02 | **0.92±0.01** |
| | Marine Easy | **1.00±0.00** | **1.00±0.00** | 0.99±0.01 | 0.82±0.09 | 0.57±0.02 | 0.99±0.00 | 0.92±0.06 | 0.99±0.01 | 0.99±0.00 |
| | Marine Hard | 0.69±0.05 | 0.69±0.01 | **0.77±0.06** | 0.46±0.10 | 0.17±0.04 | 0.53±0.20 | 0.03±0.02 | 0.62±0.04 | 0.68±0.02 |
| IID | Stalker-Zealot | **0.99±0.00** | 0.98±0.00 | 0.93±0.01 | 0.86±0.00 | 0.07±0.02 | 0.14±0.02 | 0.96±0.00 | 0.95±0.00 | **0.99±0.00** |
| | Terran | 0.52±0.01 | 0.46±0.00 | 0.43±0.03 | 0.17±0.04 | 0.06±0.01 | 0.52±0.01 | 0.52±0.04 | **0.54±0.02** | 0.50±0.02 |
| | Protoss | 0.63±0.04 | **0.66±0.02** | 0.56±0.02 | 0.27±0.06 | 0.07±0.02 | 0.60±0.01 | 0.59±0.02 | 0.65±0.01 | **0.66±0.00** |
| | Zerg | 0.54±0.02 | 0.55±0.00 | 0.50±0.01 | 0.38±0.04 | 0.29±0.00 | 0.53±0.04 | **0.56±0.01** | **0.56±0.00** | 0.55±0.03 |
| | Mean | **0.75** | 0.69 | 0.62 | 0.45 | 0.24 | 0.59 | 0.58 | 0.72 | 0.69 |
| | CN Target | **0.39±0.03** | 0.28±0.03 | 0.06±0.01 | 0.02±0.01 | 0.05±0.03 | 0.26±0.00 | 0.17±0.04 | 0.20±0.02 | 0.24±0.02 |
| | CN Agent | **0.79±0.02** | 0.77±0.01 | 0.54±0.04 | 0.33±0.17 | 0.53±0.00 | 0.71±0.02 | 0.66±0.07 | 0.69±0.00 | 0.72±0.06 |
| | Marine Easy | **0.95±0.01** | 0.92±0.01 | **0.95±0.00** | 0.56±0.14 | 0.41±0.08 | 0.88±0.04 | 0.60±0.05 | 0.93±0.01 | 0.92±0.03 |
| | Marine Hard | 0.66±0.02 | 0.55±0.02 | **0.72±0.07** | 0.27±0.05 | 0.28±0.02 | 0.45±0.15 | 0.17±0.06 | 0.47±0.11 | 0.63±0.02 |
| OOD | Stalker-Zealot | **0.72±0.00** | 0.65±0.02 | 0.56±0.04 | 0.51±0.04 | 0.06±0.02 | 0.07±0.04 | 0.55±0.01 | 0.64±0.03 | 0.67±0.01 |
| | Terran | 0.23±0.01 | 0.20±0.02 | **0.26±0.02** | 0.09±0.03 | 0.03±0.02 | 0.21±0.01 | 0.22±0.01 | 0.22±0.01 | 0.20±0.01 |
| | Protoss | **0.54±0.02** | 0.52±0.02 | 0.53±0.03 | 0.23±0.05 | 0.06±0.01 | 0.49±0.04 | 0.48±0.02 | 0.50±0.02 | 0.49±0.02 |
| | Zerg | **0.20±0.02** | 0.16±0.01 | 0.15±0.00 | 0.05±0.01 | 0.03±0.01 | 0.15±0.00 | 0.15±0.01 | 0.18±0.00 | 0.17±0.01 |
| | Mean | **0.56** | 0.51 | 0.47 | 0.26 | 0.18 | 0.40 | 0.38 | 0.48 | 0.51 |

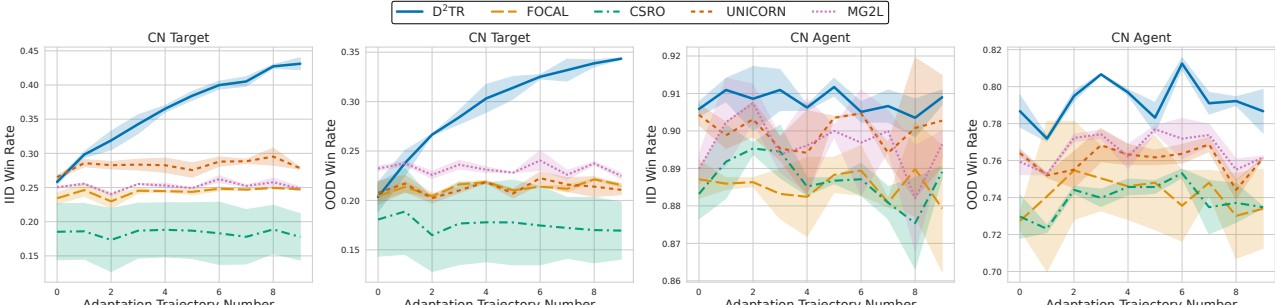

*Figure 2.* Online adaptation results.

include four multi-task baselines: **Backbone** (D$^2$TR 's backbone), **UPDeT** (Hu et al., 2021), **ODIS** (Zhang et al., 2022) and **HiSSD** (Liu et al., 2025a). Next, to validate the strong task identification and generalization capabilities of D$^2$TR's task-role encodings, we combine the Backbone with prior single-agent and multi-agent meta RL methods to form the baselines **FOCAL** (Li et al., 2021b), **CSRO** (Gao et al., 2023), **UNICORN** (Li et al., 2024), and **MG2L** (Zhao et al., 2025). Among these, FOCAL, CSRO, and UNICORN are trained directly using decentralized trajectories.

We consider the following benchmarks for evaluation.

**Cooperative Navigation (CN).** First, we conduct experiments on several CN tasks from the multi-agent particle environment (Lowe et al., 2017). To assess the performance of D$^2$TR and the baselines on tasks with different cooperative objectives, we design the **CN Target** environment. In

CN Target, there are 6 landmarks and 3 agents, with each task requiring the agents to cover a specific set of three landmarks. Since tasks have non-identical reward functions, CN Target places the greatest demand on the encoder's task identification capability and is particularly challenging for traditional multi-task algorithms. Additionally, following prior work, we also include the **CN Agent** environment, where each task has the same number of agents and landmarks, but task differences arise from varying the number of agents across tasks.

**StarCraft Multi-Agent Challenge (SMAC).** Next, to further evaluate our method on larger-scale tasks with more agents, we adopt the widely used SMAC benchmark (Whiteson et al., 2019), consistent with prior work. In SMAC, we consider three environments: **Marine Easy**, **Marine Hard**, and **Stalker–Zealot**. In Marine Easy, both sides have the same number of Marines, but the number of agents varies

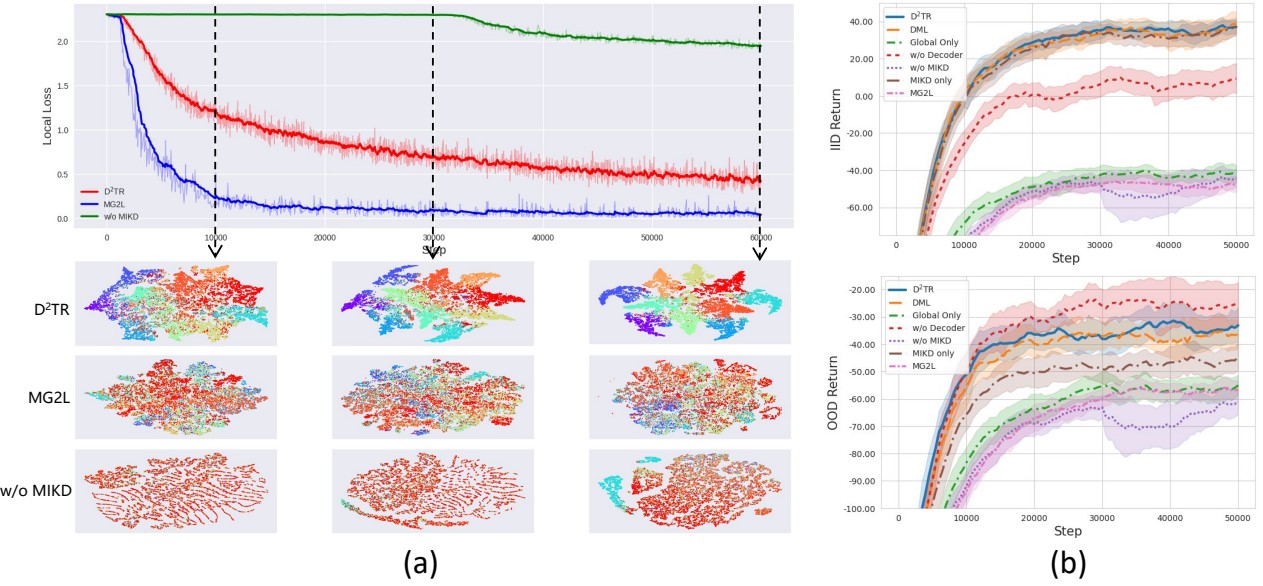

*Figure 3.* Task encoder analysis in CN Target. (a) Local encoding visualizations. (b) Smoothed ablation study results.

across tasks. In Marine Hard, the enemy team may have more units than our team. In Stalker–Zealot, both sides share the same overall unit configuration, but tasks differ in the composition of Stalkers and Zealots.

**StarCraft Multi-Agent Challenge-v2 (SMACv2).** Finally, we extend our evaluation to SMACv2, which is more complex than SMAC, and consider three environments: **Terran**, **Protoss**, and **Zerg**. Each environment involves three agent types, with the key differences across environments stemming from the specific choices of these types. Within each environment, we include 5 training tasks and 12 test tasks. For the training tasks, agent types are sampled from a uniform distribution over two of the three types, with task differences mainly arising from the selected agent types and their numbers. For the 12 test tasks, each task uses a fixed composition of all three agent types, with variations coming from agent counts and composition patterns. More details can be seen in Appendix D.

### 5.2. Competitive Results

In this section, we report the overall performance of $D^2TR$ and all baselines across all environments, as summarized in Table 1. First, Backbone consistently outperforms prior multi-task baselines, ODIS and UPDeT, across all environments in both in-distribution (IID) and out-of-distribution (OOD) settings, demonstrating the effectiveness of our actor–critic–based offline policy learning framework. While HiSSD demonstrates strong performance in Marine-related environments through more fine-grained skill learning and utilization, its overall average performance remains inferior to both the Backbone and $D^2TR$. Next, we examine

the results under the IID setting. In CN Target, which imposes stringent requirements on task identification, neither CSRO nor MG2L outperforms Backbone, despite explicitly modeling task information, indicating their ineffective encoding learning and task recognition. FOCAL and UNI-CORN achieve modest improvements, but the gains remain limited. In contrast, $D^2TR$ achieves a substantially higher win rate than all baselines, highlighting the effectiveness of the proposed mutual information knowledge distillation for task encoding learning. Even in environments where task identification is less critical, $D^2TR$ consistently matches the best-performing baselines, while avoiding the performance collapse observed in FOCAL and CSRO due to overfitting in task encoding learning. Finally, we evaluate performance in the OOD setting. None of the meta-RL baselines exhibit a clear generalization advantage over Backbone, suggesting that task encodings alone offer limited generalization in multi-agent scenarios. In contrast, $D^2TR$ consistently outperforms all baselines across all environments, highlighting the effectiveness of the proposed role-encoding design and its integration into policy learning.

To assess the robustness of policy adaptation without explicit context, we conduct an online adaptation experiment in which the policy actively collects trajectories as context. Results are averaged over three random seeds, with 20 runs per seed and up to 10 trajectories collected in each run. The results are reported in Figure 2. In the CN Target environment, $D^2TR$ exhibits clear performance gains as more adaptation trajectories are collected, achieving significantly higher win rates than all baselines in both IID and OOD settings. In CN Agent, while the advantage of $D^2TR$ is less pronounced under IID conditions, it consistently out-

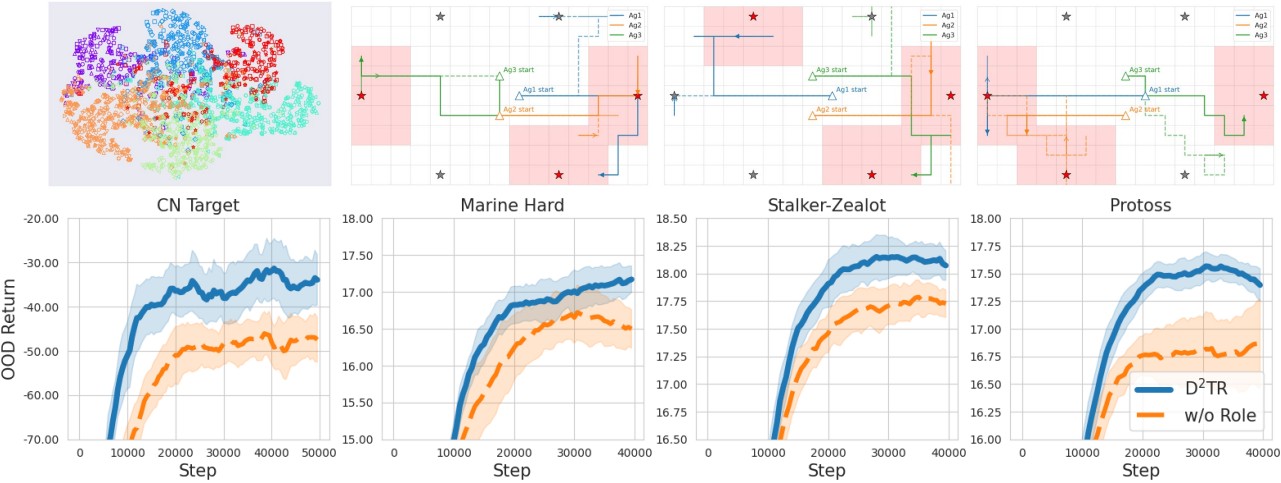

*Figure 4.* (a) Role encoding visualizations (upper left), different colors refer to different roles, while different markers refer to different tasks. (b) Trajectory visualization in CN Target (upper right). The solid lines and arrows denote the trajectories produced by $D^2$TR, while the dashed lines and hollow arrows represent the trajectories of the variant without role modeling (w/o Role). The red stars indicate the target landmarks, and the gray stars denote the remaining landmarks. (c) Ablation results (down).

performs all baselines in the OOD setting. These results demonstrate the effectiveness and robustness of $D^2$TR for online adaptation. Details of the online adaptation algorithm are provided in Appendix C, while more results are provided in Appendix E.

### 5.3. Visualization and Ablation Studies

**Task Encoder Analysis**  In this section, we aim to investigate the effectiveness of the proposed task encoder. To this end, we first visualize the training dynamics of different encoders, as shown in Figure 3(a). Comparing $D^2$TR with **w/o MIKD** (an ablation baseline that does not employ mutual information knowledge distillation), we observe that the encoder loss of $D^2$TR converges much more efficiently, dropping rapidly in the early stages of training. As a result, encodings corresponding to different tasks become clearly separable at an early stage. In contrast, for w/o MIKD, the loss only begins to decrease gradually in the later stages of training, and even in the final it can distinguish only a small subset of tasks. Next, comparing $D^2$TR with MG2L, we find that although both methods exhibit fast loss reduction, MG2L still fails to effectively separate all tasks despite the decrease in loss. This indicates that the loss reduction in MG2L mainly stems from learning the global encoding, while the information contained in the local encoding is in fact diminished. Next, we introduce additional ablation baselines for performance comparison, including **DML** (which uses a DML loss to approximate the InfoNCE loss), **Global Only** (which performs contrastive learning only on centralized trajectories and derives local encodings via a mixing-network–style structure similar to value decomposition methods), **w/o Decoder** (which removes $L_{recon}$), **w/o**

**MIKD**, and **MIKD only** (which uses only the knowledge distillation loss). The results are shown in Figure 3(b). We observe that DML achieves overall performance comparable to $D^2$TR, suggesting that replacing the InfoNCE loss with a DML loss is a viable alternative. In contrast, both Global Only and w/o MIKD suffer significant performance degradation in both IID and OOD settings similar to MG2L, demonstrating the effectiveness of our MIKD design for accurate task identification. The w/o Decoder variant shows some performance improvement in the OOD setting; however, this comes at the cost of degraded task identification and performance in the IID setting. Finally, although MIKD only maintains similar performance in the IID setting, it exhibits a noticeable performance drop in the OOD setting, highlighting the importance and effectiveness of $L_{recon}$.

**Role Encoding Analysis**  In this section, we analyze the impact of role encodings. First, we visualize the learned role encodings in the upper-left panel of Figure 4. Notably, each role encompasses multiple distinct tasks, which corroborates the rationale for introducing role encodings to facilitate knowledge transfer. Meanwhile, the role encoder clearly separates different roles, demonstrating strong discriminative capability. Next, we qualitatively compare trajectories generated by $D^2$TR and its variant without role encodings (**w/o Role**) on three OOD tasks in CN Target under the same context (upper-right panel of Figure 4). $D^2$TR consistently identifies the correct target landmarks and navigates toward them, whereas w/o Role frequently misidentifies one or more targets and exhibits behaviors biased toward IID tasks. This further confirms that task encodings alone are insufficient for robust generalization, while role encodings enable effective transfer and recombination of diverse behav-

*Figure 5.* LLM sensitivity study results.

iors. Finally, we quantitatively compare $D^2TR$ and w/o Role across four test environments by reporting average OOD returns (bottom of Figure 4). In most environments, role encodings yield clear improvements in learning efficiency or final performance, further validating the effectiveness of the proposed role encoder and its integration into policy learning. More results can be seen in Appendix E.

### 5.4. LLM Sensitivity Analysis

Finally, we analyze the sensitivity of role labeling to the choice of LLM. We first consider a code-based role labeling environment, CN Target, and a text-based role labeling environment, Stalker–Zealot, and the following baselines are then introduced: **DeepSeek V3.2**, which replaces GPT-5 with DeepSeek V3.2; **GPT-5 mini**, which replaces GPT-5 with GPT-5 mini; and **Clustering Role**, which does not use an LLM and instead assigns roles directly based on trajectory similarity via clustering. For the text-based setting, we further include two additional baselines to evaluate robustness in language processing: **New Seed**, which queries the LLM using a different random seed; and **Simple Prompt**, which removes the example phrase ("for example, a tank that lures enemies on the front line, or a kiter dealing damage from the back line") from our prompt. The results are shown in Figure 5. Overall, the conclusions are highly consistent across different environments. Both DeepSeek V3.2 and GPT-5 mini achieve performance comparable to our method across all IID and OOD settings. In the Stalker–Zealot environment, the New Seed and Simple Prompt variants also exhibit negligible performance differences, indicating that our method is robust to the choice of LLM, as well as to randomness and prompt design. In contrast, while Clustering Role achieves strong performance across all IID settings, it consistently underperforms our method in OOD scenarios. Moreover, although it surpasses the Backbone in the Stalker–Zealot environment, it performs worse than the Backbone in the CN Target environment. These results suggest that, while incorporating role embeddings can provide additional information that facilitates generalization in OOD settings, the effectiveness of this benefit is highly dependent on the quality of the role labels. Low-quality or inaccurate role assignments may not only fail to improve

generalization but can even be detrimental. Furthermore, these findings validate the effectiveness and necessity of leveraging LLMs for role labeling in our framework.

## 6. Final Remarks

In this work, we propose $D^2TR$, a novel representation learning framework for generalizable offline multi-agent meta RL. $D^2TR$ uses mutual information knowledge distillation to learn decentralized task encodings, and leverages an LLM to extract latent role labels from offline data, enabling individual-centric inference via decentralized role encodings. Empirical results across a variety of environments provide strong evidence of the effectiveness of $D^2TR$. Looking ahead, with the development of more cost-efficient and effective role label extraction methods, or by training models with larger parameter sizes on larger-scale tasks, $D^2TR$ has the potential to further enhance generalization ability in complex real-world scenarios, such as cooperative video games or embodied multi-agent tasks.

## Acknowledgements

This work was supported by the National Natural Science Foundation of China under Grants 62495090, 62495093, U23B2059, 62506159, U24A20324, the Natural Science Foundation of Jiangsu under Grants BK20241199, BK20243039, the "111 Center" (No. B26023), and Fundamental and Interdisciplinary Disciplines Breakthrough Plan of the Ministry of Education of China (No. JYB2025XDXM118). We thank the reviewers for their support and helpful discussions on improving the paper.

## Impact Statement

The goal of the work presented in this paper is to advance the development of cooperative offline meta multi-agent RL. The proposed framework is intended to enhance the generalization of agent teams, providing an effective approach for future research on open multi-agent systems. The work presented does not raise any additional ethical concerns, and thus no special discussion on ethical issues is required.

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

## A. More Related Work

**Role-assisted MARL**    Due to agent heterogeneity in multi-agent systems, effective role assignment can significantly reduce task complexity and enable more efficient team cooperation. Motivated by this insight, several prior works have introduced the concept of roles into multi-agent reinforcement learning (MARL). Representative approaches include ROMA (Wang et al., 2020) and RODE (Wang et al., 2021), which perform online role assignment through self-supervised representation learning and clustering, thereby facilitating more efficient exploration and learning. Subsequent works, such as CDS (Li et al., 2021a), adopt similar techniques to enhance the decision diversity of shared policy networks. Most of the aforementioned methods focus on role assignment at individual states. More recently, approaches such as TRAMA (Na et al., 2025) have extended this perspective by performing clustering and role assignment from a global, trajectory-level viewpoint. Nevertheless, these methods address a fundamentally different problem setting from ours. Specifically, they aim to improve decision diversity during online MARL training in order to enable efficient exploration and coordination. In contrast, our work targets the offline setting, where decision diversity is already determined by the behavior policies and cannot be influenced during learning. Our objective is to mine and exploit such diversity from offline datasets to promote knowledge transfer and skill recomposition across multiple tasks. Therefore, although we adopt the widely used QMIX algorithm for offline data collection in our experiments, combining $D^2TR$ with datasets collected using the aforementioned role-based methods holds promise for learning multi-agent policies with stronger generalization capabilities.

**LLM-assisted Decision Making**    The integration of large language models (LLMs) into reinforcement learning (RL) has emerged as a promising research direction (Cao et al., 2024), leveraging the rich semantic understanding and strong generalization capabilities of LLMs to enhance decision-making processes. One prominent line of work seeks to directly utilize LLMs to generate actions or high-level plans for decision making (Yao et al., 2023; Shinn et al., 2023; Prasad et al., 2023), while another line of research employs LLMs to produce auxiliary signals—such as rewards, goals, or constraints—to facilitate the training of RL policies (Triantafyllidis et al., 2024; Ma et al., 2025; Yan et al., 2025). Owing to their powerful representational capacity, LLMs have also been increasingly applied to multi-agent settings, where they assist with agent communication (Park et al., 2023; Liu et al., 2024; Zhang et al., 2024a), task allocation (Kannan et al., 2024), teammate modeling (Li et al., 2025), and related functionalities.

## B. Theoretical Proofs

### B.1. Proof of Theorem 4.1

First, according to the chain rule, we have

$$I(z_i^j; M, z_M) = I(z_i^j; z_M) + I(z_i^j; M|z_M) = I(z_i^j; M) + I(z_i^j; z_M|M), \tag{21}$$

which implies

$$I(z_i^j; z_M) - I(z_i^j; z_M|M) \le I(z_i^j; M) \le I(z_i^j; z_M) + H(M|z_M), \tag{22}$$

that is

$$|I(z_i^j; M) - I(z_i^j; z_M)| \le I(z_i^j; z_M|M) + H(M|z_M). \tag{23}$$

Thus, the distillation term $I(z_i^j; z_M)$ is a gap-controlled surrogate of the target objective $I(z_i^j; M)$, and the upper-gap term is reduced by maximizing $I(z_M; M)$ since $H(M|z_M) = H(M) - I(z_M; M)$ and $H(M)$ is fixed.

Meanwhile, since $z_i^j$ and $z_M$ are obtained by independently sampling trajectories from task $M$ and encoding them using the individual and global encoders, respectively, the correlation between them primarily arises from the underlying task $M$ and the encoders themselves. When the trajectories on which they are based are different and the individual and global encoders are independent of each other, we have

$$I(z_i^j; z_M|M) = 0. \tag{24}$$

Even if there exists some degree of correlation between the individual and global encoders, such correlation only affects a limited subset of the neural network bias terms when the context trajectories are different. As a result, the additional term $I(z_i^j; z_M|M)$ is limited and will not make a strong influence.

### B.2. Proof of Proposition 4.2

First, since $M \to X \to Z$ constitutes a Markov chain, by the data processing inequality, we have

$$I(Z; M) \leq I(Z; X). \tag{25}$$

Next, again due to the Markov property, we have $I(Z; M|X_t, X_b) = 0$, and thus

$$
\begin{aligned}
&I(M; X_t, X_b) - I(M; X_t, X_b|Z) \\
=&H(M) - H(M|X_t, X_b) - H(M|Z) + H(M|X_t, X_b, Z) \\
=&I(Z; M) - I(Z; M|X_t, X_b) \\
=&I(Z; M) \geq 0.
\end{aligned}
\tag{26}
$$

Therefore, we have

$$I(M; X_b) + I(M; X_t|X_b) - I(M; X_b|Z) - I(M; X_t|X_b, Z) \geq 0, \tag{27}$$

which means

$$I(M; X_b) - I(M; X_b|Z) \geq C + I(M; X_t|X_b, Z) \geq C, \tag{28}$$

where $C$ is a constant independent of $Z$. Furthermore, we have

$$
\begin{aligned}
I(Z; M) &= I(Z; M|X_b) + I(M; X_b) - I(M; X_b|Z) \\
&\geq I(Z; M|X_b) + C \\
&\equiv I(Z; M|X_b) \\
&= I(M; Z, X_t|X_b) - I(M; X_t|Z, X_b) \\
&= I(Z; M|X_t, X_b) + I(M; X_t|X_b) - I(M; X_t|Z, X_b) \\
&= C - H(X_t|Z, X_b) + H(X_t|M, Z, X_b) \\
&\equiv -H(X_t|Z, X_b) + H(X_t|M, Z, X_b) \\
&\geq -H(X_t|Z, X_b) \\
&= -H(X_t) + I(X_t; Z, X_b) \\
&\equiv I(X_t; Z, X_b) \\
&= I(Z; X_t|X_b) + I(X_t; X_b) \\
&\equiv I(Z; X_t|X_b),
\end{aligned}
\tag{29}
$$

where $\equiv$ indicates equality up to an additive constant that is independent of the optimization variables. In summary, after ignoring constants independent of $Z$, we obtain

$$I(Z; X_t|X_b) \leq I(Z; M) \leq I(Z; X_t|X_b) + I(Z; X_b) = I(Z; X). \tag{30}$$

## B.3. Proof of Theorem 4.3

This proof shares similar structure to Li et al. (2024). First, we have

$$
\begin{aligned}
I(Z;X) &:= \mathbb{E}_{x,z}\left[\log\left(\frac{p(z,x)}{p(z)p(x)}\right)\right] \\
&= \mathbb{E}_{x,z}\left[\log\left(\frac{1}{\frac{p(z)}{p(z\mid x)}|\mathcal{M}_{\text{train}}|}\right)\right] + \log(|\mathcal{M}_{\text{train}}|) \\
&= \mathbb{E}_{x,z}\left[\log\left(\frac{1}{\frac{p(z)}{p(z\mid x)}|\mathcal{M}_{\text{train}}|\,\mathbb{E}_x\left[\frac{p(z|x)}{p(z)}\right]}\right)\right] + \log(|\mathcal{M}_{\text{train}}|) \\
&\approx \mathbb{E}_{x,z}\left[\log\left(\frac{1}{\frac{p(z)}{p(z\mid x)}\sum_{M^i\in\mathcal{M}_{\text{train}}}\mathbb{E}_{x^i\sim X^i}\left[\frac{p(z|x^i)}{p(z)}\right]}\right)\right] + \log(|\mathcal{M}_{\text{train}}|) \\
&= \mathbb{E}_{x,z}\left[\log\left(\frac{\frac{p(z\mid x)}{p(z)}}{\sum_{M^i\in\mathcal{M}_{\text{train}}}\mathbb{E}_{x^i\sim X^i}\left[\frac{p(z|x^i)}{p(z)}\right]}\right)\right] + \log(|\mathcal{M}_{\text{train}}|) \\
&= \mathbb{E}_{x,z}\left[\log\left(\frac{h(x,z)}{\sum_{M^i\in\mathcal{M}_{\text{train}}}h(x^i,z)}\right)\right] + \log(|\mathcal{M}_{\text{train}}|),
\end{aligned}
\tag{31}
$$

where $h(x,z) = \frac{p(z|x)}{p(z)}$. Next, following the InfoNCE formulation (Oord et al., 2018), $h(x,z)$ can be parameterized as the exponential of a scoring function, i.e., $h(x,z) \propto \exp(S(z',z))$. Consequently, we obtain the InfoNCE loss

$$
L_{\text{InfoNCE}} = -\mathbb{E}_{x,z}\left[\log\left(\frac{\exp(S(z,z'))}{\sum_{M^i\in\mathcal{M}_{\text{train}}}\exp(S(z,z^i))}\right)\right],
\tag{32}
$$

where $z'$ denotes a positive task encoding sampled from the same task $M$, and $z^i$ denotes a negative task encoding sampled from a different task $M^i$. Therefore, we first have

$$
L_{\text{InfoNCE}} \equiv -I(Z;X).
\tag{33}
$$

It then follows that

$$
\begin{aligned}
I(Z;X_t|X_b) &= I(X_t;Z,X_b) - I(X_t;X_b) \\
&\equiv I(X_t;Z,X_b) \\
&= \mathbb{E}_{x_t,x_b,z}[\log\frac{p(x_t|z,x_b)}{p(x_t)}] \\
&\equiv \mathbb{E}_{x_t,x_b,z}[\log p(x_t|z,x_b)] \\
&\geq \mathbb{E}_{x_t,x_b,z\sim f(\cdot|x_t,x_b)}[\log p_\theta(x_t|z,x_b)],
\end{aligned}
\tag{34}
$$

where $f$ denotes the task encoder and $p_\theta$ denotes the decoder network. By defining

$$
L_{\text{recon}} = -\mathbb{E}_{x_t,x_b,z\sim f(\cdot|x_t,x_b)}[\log p_\theta(x_t|z,x_b)],
\tag{35}
$$

we can obtain that after ignoring constants independent of $Z$, $-L_{\text{recon}}$ serves as a lower bound of $I(Z;M)$ while $-L_{\text{InfoNCE}}$ provides an approximate upper bound of $I(Z;M)$. Therefore, by forming a convex combination of the two, we obtain an approximate representation of $I(Z;M)$

$$
I(Z;M) \approx -\beta L_{\text{InfoNCE}} - (1-\beta)L_{\text{recon}},
\tag{36}
$$

where $\beta \in [0, 1]$.

## C. Implementation Details

### C.1. Network Architecture Design

First, in multi-agent settings, different tasks may involve varying numbers of allies and enemies. Therefore, similar to UPDeT and ODIS, we employ Transformer networks to handle variable-length observations and actions. For an observation $o$, we first decompose it into the following components:

$$o = \left(o_{\text{self}}, o_{\text{ally}_1}, \ldots, o_{\text{ally}_{n_{\text{ally}}}}, o_{\text{enemy}_1}, \ldots, o_{\text{enemy}_{n_{\text{enemy}}}}\right), \tag{37}$$

where $n_{\text{ally}}$ refers to ally number and $n_{\text{enemy}}$ refers to enemy number. Next, for an action $a$ (one-hot), we similarly decompose it as follows:

$$a = \left(a_{\text{move}}, a_{\text{attack/heal}_1}, \ldots, a_{\text{attack/heal}_{n_a}}\right), \tag{38}$$

where $n_a$ refers to the agent number that can be affected by $a$. Next, when the action $a$ needs to be used as an input to the neural network, we concatenate all components except $a_{\text{move}}$ to the corresponding agent-specific part of the observation. These concatenated components are then ignored in subsequent processing. After incorporating the local encoding $z^i$ and the role encoding $z_r^i$, we obtain the following input sequence:

$$x = \left(z^i, z_r^i, o_{\text{self}}, o_{\text{ally}_1}, \ldots, o_{\text{ally}_{n_{\text{ally}}}}, o_{\text{enemy}_1}, \ldots, o_{\text{enemy}_{n_{\text{enemy}}}}\right). \tag{39}$$

Then, each component of the input $x$ is processed by its corresponding one-layer MLP tokenization network and transformed into tokens with the same dimensionality. Different tokenization networks are used for different encodings, including the local encodings, $o_{\text{self}}$, ally observations, and enemy observations. Meanwhile, allies (or enemies) with different indices share the same tokenization network. For simplicity, we denote all tokenization networks collectively as $T$. Furthermore, since allies (or enemies) with different indices are processed by the same network, an additional ID embedding network $\mathcal{E}$ is introduced to distinguish them. As a result, the tokenized input after processing, denoted as $Tx$, can be expressed as

$$Tx = (T(z^i), T(z_r^i), T(o_{\text{self}}), T(o_{\text{ally}_1}) + \mathcal{E}(1), \ldots, T(o_{\text{ally}_{n_{\text{ally}}}}) + \mathcal{E}(n_{\text{ally}}), T(o_{\text{enemy}_1}) + \mathcal{E}(1), \ldots, T(o_{\text{enemy}_{n_{\text{enemy}}}}) + \mathcal{E}(n_{\text{enemy}})). \tag{40}$$

Subsequently, $Tx$ is processed by the Transformer network to produce the sequence output $y$

$$y = (y^i, y_r^i, y_{\text{self}}, y_{\text{ally}_1}, \ldots, y_{\text{ally}_{n_{\text{ally}}}}, y_{\text{enemy}_1}, \ldots, y_{\text{enemy}_{n_{\text{enemy}}}}). \tag{41}$$

Next, when the output is used to generate the task-invariant action $a_{\text{move}}$, we concatenate $y^i, y_r^i, y_{\text{self}}$ with the max-pooled representations over $\{y_{\text{ally}_1}, \ldots, y_{\text{ally}_{n_{\text{ally}}}}\}$ and $\{y_{\text{enemy}_1}, \ldots, y_{\text{enemy}_{n_{\text{enemy}}}}\}$, and feed the resulting vector into a subsequent MLP. In contrast, when generating attack or heal actions, we directly use the Transformer output corresponding to the selected ally or enemy position as the input to the subsequent MLP network. Moreover, we share parameters for decentralized policies and value networks across agents and tasks.

For the task encoder and the role encoder networks, we similarly process the transition $(o, a)$ using the aforementioned procedure. The key difference is that the task encoder does not take any encoding as input, whereas the role encoder only incorporates the local encoding $z^i$ as auxiliary input for learning. The remaining processing steps are identical to those used for generating $a_{\text{move}}$. After processing at the transition level, we aggregate the representations of different transitions within the context to obtain $y_{\text{transition}}^i = (y_1^i, \ldots, y_t^i, \ldots)$, where $y_t^i$ refers to the processed output of the t-th transition. For this aggregated representation, we further apply a gated mean operation to integrate information along the temporal dimension

$$y_{\text{temporal}}^i = \sum_t \text{Softmax}(G(y_1^i), \ldots, G(y_t^i), \ldots)_t V(y_i^t), \tag{42}$$

where $G$ and $V$ are two linear networks. Subsequently, $y_{\text{temporal}}^i$ is passed through several MLP layers to produce either the local encoding $z^i$ or the role encoding $z_r^i$. Finally, for the global encoding $z$, rather than introducing an additional dedicated network, we directly feed the local encodings of all agents $[z^1, z^2, \ldots, z^n]$ into a Transformer network to aggregate them, thereby enabling more effective knowledge sharing.

## C.2. Trajectory Pre-clustering

In $D^2TR$, for tasks without explicit subgoals, it is necessary to perform pre-clustering on trajectories before applying the LLM for role assignment. In principle, an alternative approach would be to directly use the LLM to describe all trajectories and then perform clustering or merging based on the generated descriptions. Such an approach could potentially yield stronger semantic completeness, since pre-clustering—being purely data-driven—may inevitably produce clusters that are not fully aligned with semantic distinctions. Nevertheless, we adopt pre-clustering primarily due to the substantial reduction in computational cost it provides. After pre-clustering, the total number of decentralized trajectories that require LLM-based description is $K \times T \times E$. In the largest configuration used in our experiments, $(K = 6, T = 5, E = 10)$, this amounts to only 300 trajectories. In contrast, without pre-clustering, even when considering a single training task with 10 allied agents, 4,000 centralized trajectories would correspond to 40,000 decentralized trajectories requiring LLM-based description—over two orders of magnitude larger than with pre-clustering. Therefore, although $D^2TR$ does not impose specific requirements on the choice of pre-clustering method, employing pre-clustering is essential in practice due to its significant advantages in reducing computational and annotation costs.

In our experiments, we employ the trajectory clustering method used in TRAMA (Na et al., 2025). Briefly, this method first encodes all observations using a VQ-VAE (van den Oord et al., 2017). For a given decentralized trajectory, the method then uses the sum of the quantized vectors of all observations in the trajectory as its representation. This representation is subsequently used in a K-Means clustering algorithm to perform the clustering. In $D^2TR$, however, other simple and effective pre-clustering methods can be readily substituted.

## C.3. Prompt Engineering for LLM-assisted Role Labeling

Since prompt design is crucial for LLM-based role labeling, we present the prompt used in this work in this section. All of these prompts are processed using **GPT-5** as the underlying LLM.

First, taking the CN Target environment as an example, we introduce the prompt design for LLM-based code generation in environments with explicit subgoals. First, we provide task-related information

---

**Task**

The agents are playing the cooperative navigation game. In this game, there are 3 agents and 6 landmarks. In the current task, the {task_target} landmarks are target landmarks, if all the target landmarks are reached by the agents, the task is considered successful. If the x-coordinate distance between the agent and the landmark is smaller than 3, and the y-coordinate distance between the agent and the landmark is smaller than 3, then the agent is considered to have reached the landmark.

The observation is formulated as [agent_x, agent_y, ally_1_x, ally_1_y, ally_2_x, ally_2_y, target_0_x, target_0_y, target_1_x, target_1_y, target_2_x, target_2_y, target_3_x, target_3_y, target_4_x, target_4_y, target_5_x, target_5_y].

The trajectory is a sequence (list) of observations.

---

Next, we construct the first instruction to prompt the LLM to describe the behaviors exhibited in the trajectories

---

**Instruction1**

Based on the given trajectory, please provide a description of the given agent's behavioral objective. For example, for the trajectory {str(example traj)}, the agent tried to near landmarks {landmark ID} in target landmarks, and finally (in the last step) reached landmark {landmark ID}.

Please describe the behavioral objectives of the given agent in the trajectory {str(target traj)}. Before output the final answer, please check twice and think step by step.

---

Using the task prompt together with the aforementioned instruction, we obtain a description of the behaviors exhibited in the trajectory. This behavioral description is then provided as input to the LLM, along with the following instruction, to produce a concise summary of the role associated with the trajectory

---

**Instruction2**

Based on the analysis above, please summarize the role of the given agent in a short phrase. The output format should be: Role:_.

---

After applying the above procedure to the decentralized trajectory of each agent within a global trajectory, we obtain a role description for every agent. We then unify the role descriptions across all agents for subsequent processing

---

**Instruction3**

Based on the previous analysis that {previous response}, please unify the format of the roles for each agent. The output format should be: 0:role_0, 1:role_1, 2:role_2.

---

Based on this procedure, we obtain a role summary for a fixed global trajectory under a given task. We then apply this summarization process to $E$ sampled trajectories for each task and aggregate all resulting summaries. This aggregation yields a document containing diverse role descriptions. By further refining and distilling this document, we ultimately derive a comprehensive list of all possible roles

---

**Instruction4**

In the current environment, the following role list has been summarized {role list}, where many items have similar meanings. Please merge the items with similar meanings in this role list to obtain the final simplified role list. The output format should be: role ls: _.

---

Finally, based on the derived role list, we use the task prompt together with the following instruction to generate code with the LLM

---

**Instruction 5**

Given the role list {role list}, please output a function code to choose one of the role in the summarized role list (if there are multiple reasonable choises, pick one of them). The input of the function includes the trajecotry and the target landmark index list, and the output of the function is the role index. Before output the final code, please check twice and think step by step.

---

The generated code is

---

**Output Code**

```python
def choose_role_index(trajectory, target_landmark_indices):
    """
    Choose the role index based on the agent's final position and the target
    ↪  landmarks.

    Args:
        trajectory: List of observations (each observation is a list of 18 floats).
        target_landmark_indices: List of indices of target landmarks (e.g., [0, 1,
        ↪  2]).

    Returns:
        Index (int) in the target_landmark_indices list that the agent was trying to
        ↪  reach.
    """
    final_obs = trajectory[-1]
    agent_x, agent_y = final_obs[0], final_obs[1]

    reached_landmarks = []
```

```
    for i, lm_idx in enumerate(target_landmark_indices):
        lm_x = final_obs[6 + 2 * lm_idx]
        lm_y = final_obs[6 + 2 * lm_idx + 1]

        if abs(agent_x - lm_x) < 3 and abs(agent_y - lm_y) < 3:
            # Compute Euclidean distance for tie-breaking
            dist = ((agent_x - lm_x) ** 2 + (agent_y - lm_y) ** 2) ** 0.5
            reached_landmarks.append((i, dist))

    if not reached_landmarks:
        # If no landmark is reached, return the closest one
        min_dist = float('inf')
        best_idx = 0
        for i, lm_idx in enumerate(target_landmark_indices):
            lm_x = final_obs[6 + 2 * lm_idx]
            lm_y = final_obs[6 + 2 * lm_idx + 1]
            dist = ((agent_x - lm_x) ** 2 + (agent_y - lm_y) ** 2) ** 0.5
            if dist < min_dist:
                min_dist = dist
                best_idx = i
        return target_landmark_indices[best_idx]
    else:
        # Return the index of the closest reached landmark
        reached_landmarks.sort(key=lambda x: x[1])
        return target_landmark_indices[reached_landmarks[0][0]]
```

Then, taking the Terran environment as an example, we introduce the prompt design for LLM-based description and merging in environments without explicit subgoals. In this environment, since vectorized observations and actions are preprocessed into textual descriptions using code, the task prompt is relatively simple

**Task**

The agents are playing the task map of the SMAC multi-agent environment. In this game, the agents need to cooperate with allies to beat all the enemies.

Next, we directly prompt the LLM to describe the textualized trajectories

**Instruction1**

Following, you are an expert SMAC player. You will be given the trajectory of one of the agents. Next, you need to provide a concise description of that trajectory; in your description you should highlight the agent's behavioral style and characteristics, and, where possible, summarize the role the agent played — for example, a tank that lures enemies on the front line, or a kiter dealing damage from the back line. Each trajectory is made up of multiple steps, and each step contains two quantities: Observation and Action. Please respond after careful consideration, your final answer should be within <answer>and </answer>tags. Below is the trajectory:

After obtaining descriptions for the $E$ trajectories within a cluster, we summarize them with

**Instruction2**

Following, you are an expert SMAC player. You will be given several descriptive summaries of typical trajectories belonging to a specific type of agent. Please analyze and synthesize these descriptions to summarize the role this agent typically plays within the team. Please respond after careful consideration, your final answer should be within <answer>and </answer>tags. Below are the descriptions:

Finally, we merge different clusters based on their descriptions to obtain the final role assignments

**Instruction3**

Following, you are an expert SMAC player. Across multiple tasks, we have roughly categorized the roles that agents play within SMAC teams into several types, and for each type, a linguistic description of its behavioral style and role has been generated. However, many of these role types are actually identical in essence. Based on the linguistic descriptions of each role type, please merge those that represent the same underlying role, and output the merged role categories along with the list of original category names included within each merged category. Be cautious in merging—if two classes show a noticeable degree of difference, keep them separate. The final number of merged role categories can be between 5 and 9. Below, we will provide several class names along with their corresponding language descriptions in the format Class Name: , Description: ;. Please analyze the semantic content of each description and merge the classes that represent the same underlying role type. Use 0, 1, 2, ... to denote the merged role categories, and output your results strictly in the following format: <answer>0:{Class Name1, Class Name2, ... }, 1:{Class Name3, Class Name4, ... }, ... </answer>. Please respond after careful consideration. Below are the class names and descriptions:

Following, we present the LLM-generated description summaries for six clusters in the MarineMarauder_5v5 task of the Terran environment

**Description Summary in MarineMarauder_5v5 Task**

"0": "The agent is a Terran Marine that functions as a steady mid/backline DPS anchor. It typically advances with or just behind the Marauder frontline, then plants to deliver sustained, coordinated focus fire—prioritizing weakened enemy Marines first and shifting to Marauders as targets present. Micro is minimal: it favors stand-and-shoot uptime over kiting, trading HP when needed to keep pressure and secure finishes, with only measured retreats or lateral adjustments to maintain range or re-enter after damage. Overall, it's a reliable damage-throughput and finisher piece that syncs with allied focus, helps collapse low-HP targets, and anchors the firefight rather than scouting or hard-tanking.",

"1": "The agent typically plays a frontline/midline DPS anchor—more bruiser than skirmisher. Whether spawning as a Marine or Marauder, it advances to an angle, holds ground, and tunnel-visions priority targets (often enemy Marauders), trading its own HP to keep continuous focus-fire pressure and secure picks. Micro is minimal: little kiting, stutter-step, or retreat, and limited situational awareness or coordination. It draws aggro and stabilizes a lane by acting as a pressure anchor that finishes targets and then rotates to the next. Occasional exceptions include brief safety retreats or a flanky hit-and-run on the edge, but the dominant pattern is steadfast, target-fixated, trade-heavy DPS that leverages HP as a resource rather than peeling or actively protecting allies.",

"2": "The agent is a Marauder that serves as a forward-anchored bruiser/anchor for the team. It advances to a central or slightly forward lane and holds ground, trading health to maintain pressure rather than flanking or extensive kiting. Its combat pattern is disciplined focus fire: prioritizing nearby or low-HP targets (often finishing exposed Marines) while steadily pressuring enemy Marauders, with high target persistence and opportunistic kill-secures. Micro is measured—mostly stationary with occasional stutter-steps and brief retreats only when at critical health. Functionally, it draws aggro, soaks damage, and controls space to let allied Marines/Marauders fire safely, while providing reliable, sustained DPS that helps secure key takedowns and close out fights.",

"3": "The agent is a steady midline anchor/bruiser DPS. It advances with the team, holds formation near the front, and commits to disciplined focus fire—typically thinning nearby/low-HP enemy Marines first, then pivoting to enemy Marauders (leveraging anti-armor when it is a Marauder). Micro is minimal: brief lateral steps or light stutter-step to maintain range, but little kiting or retreating. It is highly commitment-oriented, trading its own HP to keep guns firing, often drawing enemy attention so allied Marines can shoot safely and helping to finish exposed targets. While not a true hard tank or a backline kiter, it functions as the team's pressure anchor/vanguard, absorbing substantial damage, coordinating implicitly with ally focus fire, and providing sustained single-target DPS. Occasionally it peels to a flank and acts as soft bait, but its typical impact comes from anchoring the midline and securing kills under fire.",

"4": " Typical role: frontline bruiser/off-tank anchor and finisher. Synthesis:- Primary identity is a forward anchor—usually a Marauder but similarly played as a Marine—who initiates or meets first contact, stands on the mid/front line, and trades HP for pressure.- Consistently draws enemy aggro early, absorbing heavy damage (often to single-digit HP) to create space for allies while delivering steady midline DPS.- Target behavior is focus-fire oriented:

> prioritizes exposed/high-value Marauders or weakest enemies, and as Marine often cleans up low-HP targets to secure kills.- Micro pattern is advance-and-shoot with minimal kiting: frequent short stutter steps or brief retreats under pressure, but rarely sustained kiting or disengage; tends to be target-fixated.- Team impact: enables allied DPS by anchoring the fight and forcing enemy attention; pressures priority targets and helps convert kills, at the cost of survivability and occasional overextension/isolation.- Outliers exist (rare cautious/avoidant paths that isolate and contribute little), but the dominant trend is an aggressive, committed frontliner that trades durability for tempo and kill pressure.",
>
> "5": "- Core identity: a midline-to-frontline bruiser/anchor that trades space for pressure, whether spawned as a Marauder or a Marine.- Positioning: advances with the team to a central/right-front firing line, then largely holds ground with minimal kiting; retreats only when near-critical or to re-enter opportunistically.- Targeting priorities: disciplined focus fire on high-impact threats first (enemy Marauders), then nearby/low-HP Marines; strong finisher instincts to convert picks once targets are chipped down.- Micro style: low APM, stand-and-trade; relies on range discipline enough to land steady volleys but can exhibit target fixation and occasional out-of-vision/out-of-range attacks.- Team impact: soaks meaningful damage and draws aggro, stabilizes the front, and provides consistent DPS that enables allies to follow up; often secures kills to collapse the fight rather than peeling or hard flanking.- What it is not: not a pure backline kiter or flanker; plays more as a line holder/brawler with selective disengages rather than continuous kite micro."

### C.4. Prior Role Encoder for Online Adaptation

The role encoder is designed to extract role information from a given context and make decisions based on the inferred roles. However, during online adaptation, agents are required to actively explore the environment, and the trajectories collected during exploration tend to exhibit a high degree of randomness. As a result, the role information inferred from such trajectories may be suboptimal. Consequently, directly relying on the role encoder to make decisions based on these contexts can adversely affect online adaptation performance.

To address this issue, we introduce an additional prior role encoder, denoted as $f_{PR}$, which is trained jointly with the role encoder. The prior role encoder takes the local task encoding $z^i$ and the observation $o_t$ as inputs, and is optimized using the following loss function

$$\mathcal{L}_{\text{prior}} = \mathbb{E}_{x_b \sim \mathcal{D}, o_t \sim x_b, z^i} \left[ (f_{PR}(z^i, o_t^i) - f_R(z^i, x_b))^2 \right], \tag{43}$$

where $z^i$ denotes the local task encoding corresponding to the task of $x_b$, which is incorporated into the role encoder to enrich the input information and facilitate learning. With the prior role encoder in place, during the first $H$ exploration episodes of online adaptation, we use the prior role encoder to generate role encodings. After the initial $H$ episodes, role encodings are instead produced by the role encoder using the best-performing trajectories collected during exploration.

### C.5. Pseudo-codes

In this section, we provide the detailed pseudo-codes for D$^2$TR's training (Algorithm 1) and adaptation (Algorithm 2).

### C.6. Hyperparameters

Finally, we provide the hyperparameters used during training to ensure reproducibility, as shown in Table 2. Specifically, for learning epoch, we set it to 60000 for encoder learning and 50000 for policy learning in CN tasks, while 20000 for encoder learning and 40000 for policy learning in SMAC and SMACv2 tasks. For cluster num $K$, we set it to 4 in SMAC tasks, while 6 in SMACv2 tasks. D$^2$TR is implemented based on Offpymarl [1] and ODIS [2] code bases, and the default parameters are retained for any hyperparameters not explicitly mentioned.

## D. Detailed Description of the Baselines and Tasks

### D.1. Baselines

We provide a more detailed introduction to the baselines in this section.

---

[1]https://github.com/zzq-bot/offline-marl-framework-offpymarl
[2]https://github.com/LAMDA-RL/ODIS

---

**Algorithm 1 $\mathtt{D}^2\mathtt{TR}$ Offline Training**

---

**Input:** offline multi-task dataset $\mathcal{D}$, LLM generated role labels.
**Initialize:** individual task encoder $f_I$, global task encoder $f_G$, role encoder $f_R$, prior role encoder $f_{PR}$, individual critic $Q_i$, individual value $V_i$, mixing network parameter $\omega_i, b$, decentralized policy $\pi_i$, $i = 1, \ldots, n$.
**for** step in task encoder training steps **do**
    Update $f_G$ with Equation (8).
    Update $f_I$ with Equation (9).
**end for**
**for** step in role encoder training steps **do**
    Update $f_R$ with Equation (16).
    Update $f_{PR}$ with Equation (43).
**end for**
**for** step in policy training steps **do**
    Update $V_i$ with Equation (17).
    Update $Q_i, \omega_i, b$ with Equation (18).
    Update $\pi_i$ with Equation (19).
**end for**
Return $f_I, f_G, f_R, f_{PR}, Q_i, V_i, \omega_i, b, \pi_i, i = 1, \ldots, n$.

---

---

**Algorithm 2 $\mathtt{D}^2\mathtt{TR}$ Online Adaptation**

---

**Input:** test task $M$, individual task encoder $f_I$, role encoder $f_R$, prior role encoder $f_{PR}$, decentralized policy $\pi_i, i = 1 \ldots, n$, prior role adaptation episode $H$.
**Initialize:** contexts $c^i = \emptyset, i = 1, \ldots, n$.
**for** adaptation episode $h$ in $1, \ldots, h_{\max}$ **do**
    Obtain $z^i = f_I(c^i), i = 1 \ldots, n$.
    **if** $h < H$ **then**
        Obtain $z_r^i$ using $f_{PR}, i = 1, \ldots, n$.
    **else**
        Obtain $z_r^i$ using $f_R$ with the best-performing trajectory in $c^i, i = 1, \ldots, n$.
    **end if**
    Rollout policy $\prod_{i=1}^n \pi_i(\cdot | o^i, z^i, z_r^i)$ in $M$ and store trajectories into $c^i, i = 1, \ldots, n$.
**end for**

---

- **UPDeT** is a classical multi-task MARL framework. It is among the first to decompose observations into self, ally, and enemy components, enabling the use of Transformer-based architectures to process variable-length observation spaces in a sequential manner, facilitating more efficient multi-task learning and knowledge transfer across tasks.

- **ODIS** is another widely adopted baseline in multi-task MARL. It replaces the original agent action space with a skill space, with the objective of leveraging the generality of learned skills to achieve improved generalization performance when facing OOD tasks.

- **FOCAL** is one of the most representative methods in context-based offline meta RL. It employs a DML loss to pull together the encodings of the same task while pushing apart those of different tasks, thereby learning a context encoder with strong representational capacity.

- **CSRO** is a recently proposed and effective method in context-based offline meta RL. By introducing the CLUB loss, it explicitly decouples the context encoder from the behavior policy, thereby enabling the learned policy to robustly identify tasks from contexts collected with varying data quality and under different behavior policies.

- **UNICORN** is a state-of-the-art method in context-based offline meta RL. From a theoretical perspective, it combines the DML loss proposed in FOCAL with a reconstruction loss to maximize the mutual information between task representations and their encodings. As a result, UNICORN demonstrates robust and efficient task identification as well as strong generalization capability.

*Table 2.* Hyperparameter choices of $D^2TR$.

|  | Hyperparameter | Value |
|---|---|---|
| task and role encoder learning | Transformer token dim | 32 |
|  | Transformer head | 1 |
|  | Transformer depth | 1 |
|  | output encoding dim | 16 |
|  | meta batch size | 16 |
|  | learning rate | 0.001 |
|  | learning epoch | $\{60000, 20000\}$ |
|  | $\alpha$ | 1.0 |
|  | $\beta$ | 0.9 |
| role labeling | cluster num $K$ | $\{4, 6\}$ |
|  | example trajectory num $E$ | 10 |
| online adaptation | prior role adaptation num $H$ | 5 |
| policy learning | Transformer token dim | 64 |
|  | Transformer head | 8 |
|  | Transformer depth | 3 |
|  | trajectory batch size | $\{32, 2\}$ |
|  | actor learning rate | $5e-4$ |
|  | critic learning rate | $5e-4$ |
|  | learning epoch | $\{50000, 40000\}$ |
|  | OMIGA temperature $\epsilon$ | 10 |
|  | $\gamma$ | 0.99 |
|  | soft update $\alpha$ | 0.005 |

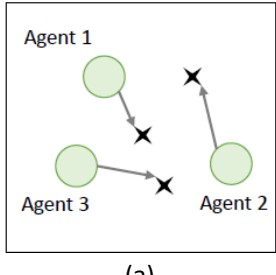 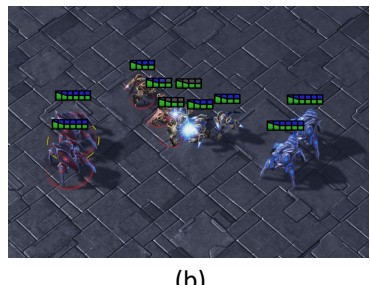 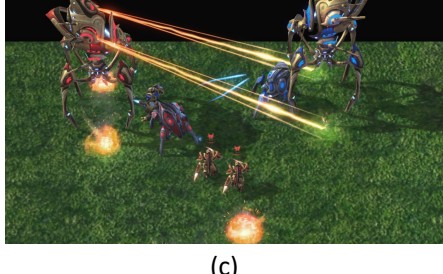

(a)          (b)          (c)

*Figure 6.* Illustrations of environments. (a) Cooperative Navigation. (b) SMAC. (c) SMACv2.

## D.2. Environments and Tasks

In this section, we provide a more detailed description of the environments and corresponding tasks involved in our experiments. Visualizations of several environments are shown in Figure 6.

**CN Target**   The CN Target environment is adapted from the CN environment in Grid MPE. In this environment, there are three cooperative agents and six landmarks, among which three are designated as target landmarks. The agents' objective is to cover all three target landmarks; coverage is considered successful when the absolute distances along both the $x-$ and $y-$axes between an agent and a target landmark are $\leq 2$. An agent receives a reward whenever its distance to the nearest target landmark decreases. The global reward is defined as the sum of the proximity rewards of the three agents and an additional terminal reward indicating whether successful coverage has been achieved. By selecting three target landmarks from the six available landmarks, we obtain $\binom{6}{3} = 20$ distinct tasks. Among them, 10 tasks are used for training and the remaining 10 for evaluation. All tasks are indexed using integers from 0 to 19.

**CN Agent**    The CN Agent environment is constructed in a manner similar to CN Target. The key difference is that the number of landmarks is equal to the number of agents, and all landmarks are treated as target landmarks. The variations across tasks primarily arise from differences in the number of agents. Specifically, tasks with 2 agents and 4 agents are used for training, while tasks with 3 agents and 5 agents are reserved for evaluation.

**Marine Easy**    The Marine Easy environment involves only a single unit type, namely Marine, with the same number of agents on both the allied and enemy sides, ensuring a symmetric setting. Tasks are named according to the number of agents; for example, 3m denotes a scenario in which both sides consist of three Marine agents. In this environment, the training tasks include {3m,5m,7m,10m}, while the test tasks consist of {4m,6m,8m,9m,11m,12m}.

**Marine Hard**    The Marine Hard environment also involves a single unit type, Marine, but the numbers of allied and enemy agents are not necessarily symmetric. Tasks are named using both the allied and enemy agent counts; for example, 5m_vs_6m denotes a scenario with five allied Marine agents facing six enemy Marine agents.  In this environment, the training tasks include {3m,5m_vs_6m,9m_vs_10m,10m_vs_12m} while the test tasks consist of {4m,5m,7m_vs_8m,8m_vs_9m,10m_vs_11m,13m_vs_15m}.

**Stalker-Zealot**    All tasks in the Stalker–Zealot environment consist of two unit types, Stalker and Zealot, with symmetric compositions between the allied and enemy sides. Tasks are named according to the numbers of Stalker and Zealot units; for example, 2s3z denotes a scenario in which both sides contain two Stalker agents and three Zealot agents. The training tasks include {2s3z, 2s4z, 3s5z}, while the test tasks consist of {1s3z, 1s4z, 1s5z, 2s5z, 3s3z, 3s4z, 4s3z, 4s4z, 4s5z}.

**Terran**    Terran is an environment in SMACv2, primarily composed of three unit types: Marine, Marauder, and Medivac.  The numbers and compositions of allied and enemy agents are not necessarily symmetric. Notably, the construction of training and test tasks in SMACv2 differs from that of the original SMAC environment.  For all training tasks, each scenario consists of only two of the three unit types.  The number of each unit type is not fixed but sampled from a uniform distribution.  When the numbers of allied and enemy agents are equal, their compositions are symmetric.  Otherwise, if the enemy side contains more agents, the overlapping portion shares the same composition as the allied side, while the additional enemy agents are sampled independently according to the same distribution.  For example, in the MarineMarauder_5v5 task, each allied agent has a 0.5 probability of being a Marine and a 0.5 probability of being a Marauder, with the enemy side mirroring the same composition.  In contrast, in the MarineMarauder_10v11 task, the first 10 enemy agents have the same composition as the allied agents, while the 11th enemy agent is independently sampled from the uniform distribution.  The training tasks in this environment include {MarineMarauder_5v5,MarineMedivac_5v5,MarauderMedivac_10v10,MarineMarauder_10v11,MarineMedivac_10v11}. For the test tasks, agent compositions are no longer stochastic and are instead explicitly specified by the number of each unit type.  For example, 1m3m1m denotes a composition of one Marine, three Marauders, and one Medivac. The complete set of test tasks is {1m3m1m,2m2m1m,3m1m1m,3m3m4m,1m5m4m,5m1m4m,5m3m2m_vs_6m3m2m,2m6m2m_vs_3m6m2m, 4m3m3m_vs_5m3m3m,8m2m10m,6m9m5m,5m8m7m}.

**Protoss**    The Protoss environment primarily involves three unit types: Stalker, Zealot, and Colossus. The naming and construction of training and test tasks follow the same logic as in the Terran environment.  The training tasks include {StalkerZealot_5v5,StalkerColossus_5v5,ZealotColossus_10v10,StalkerZealot_10v11,StalkerColossus_10v11}, while the test tasks consist of {1s3z1c,2s2z1c,1s1z3c,4s3z3c,5s2z3c,2s5z3c,3s2z5c_vs_3s3z5c,2s7z1c_vs_2s7z2c,5s4z1c_vs_6s4z1c, 5s6z9c,5s8z7c,8s7z5c}.

**Zerg**    The Zerg environment primarily involves three unit types: Zergling, Hydralisk, and Baneling. The naming and construction of training and test tasks follow the same logic as in the Terran environment. The training tasks include {ZerglingHydralisk_5v5,ZerglingBaneling_5v5,HydraliskBaneling_10v10,ZerglingHydralisk_10v11,ZerglingBaneling_10v11}, while the test tasks consist of {2z1h2b,1z3h1b,3z1h1b,3z5h2b,4z3h3b,4z1h5b,4z2h4b_vs_5z2h4b,2z4h4b_vs_2z5h4b, 3z4h3b_vs_3z4h4b,10z3h7b,7z11h2b,5z6h9b}.

### D.3. Offline Dataset

Since the performance of the algorithms is highly influenced by the quality of the offline datasets, we also provide details of the offline data in this section, including the number of trajectories used for each training task, the average win rate on the

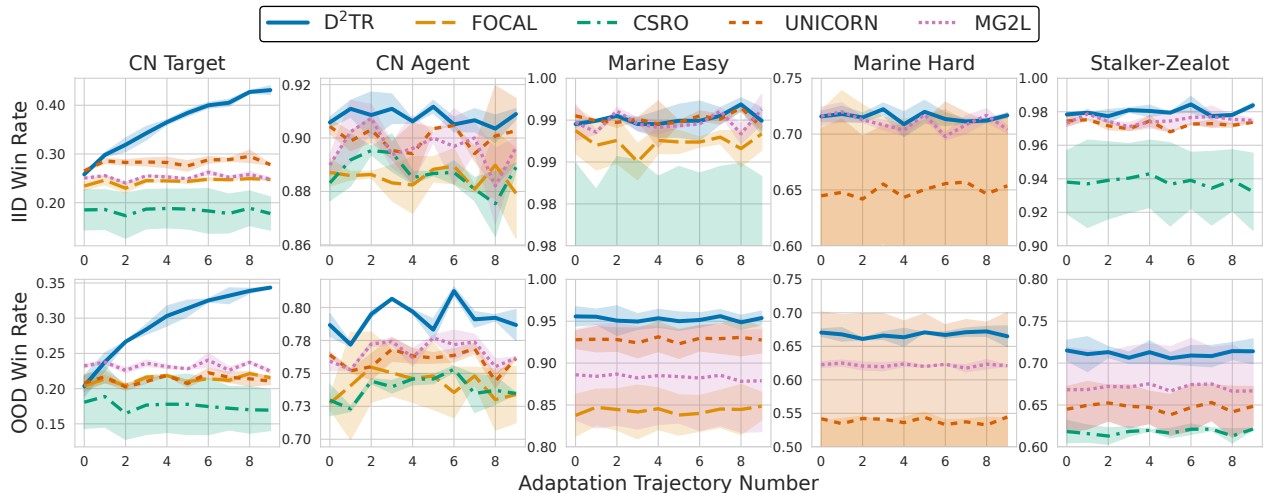

*Figure 7.* More online adaptation results.

training tasks, and the average win rate on the test tasks, as shown in Table 3.

*Table 3.* Details of the offline dataset.

| Environment | Data Num Per Train Task | Average Train Task Win Rate | Average Test Task Win Rate |
|---|---|---|---|
| CN Target | 2000 | 0.87 | 0.80 |
| CN Agent | 2000 | 0.92 | 0.72 |
| Marine Easy | 4000 | 0.99 | 0.91 |
| Marine Hard | 4000 | 0.59 | 0.79 |
| Stalker-Zealot | 4000 | 0.98 | 0.95 |
| Terran | 8000 | 0.60 | 0.56 |
| Protoss | 8000 | 0.65 | 0.58 |
| Zerg | 8000 | 0.57 | 0.44 |

# E. More Experimental Results

### E.1. Time Complexity Analysis

To better understand the practical utility of D$^2$TR, we evaluate its training time overhead in this section. Using the Stalker-Zealot environment as an example, we report the training time measured on a single NVIDIA 4090 GPU under an otherwise idle setting (i.e., with no additional CPU/GPU workload), as summarized in Table 4. It can be observed that the overall computational cost is still dominated by policy training, while the additional overhead introduced by our method—such as labeling and encoder training—remains within an acceptable range.

*Table 4.* Time complexity analysis.

| Task Encoder Training | Pre-clustering | Role Labeling | Role Encoder Training | Policy Training |
|---|---|---|---|---|
| 1.67h/task | 1.09h/task | 0.45h/task | 1.17h/task | 6.31h/task |

### E.2. More Online Adaptation Results

In this section, we further evaluate online adaptation in three additional environments. The results are shown in Figure 7 (curves not shown correspond to performance below the minimum value on the y-axis). Consistent with the findings in the main paper, D$^2$TR demonstrates more robust adaptation capabilities under both IID and OOD settings.

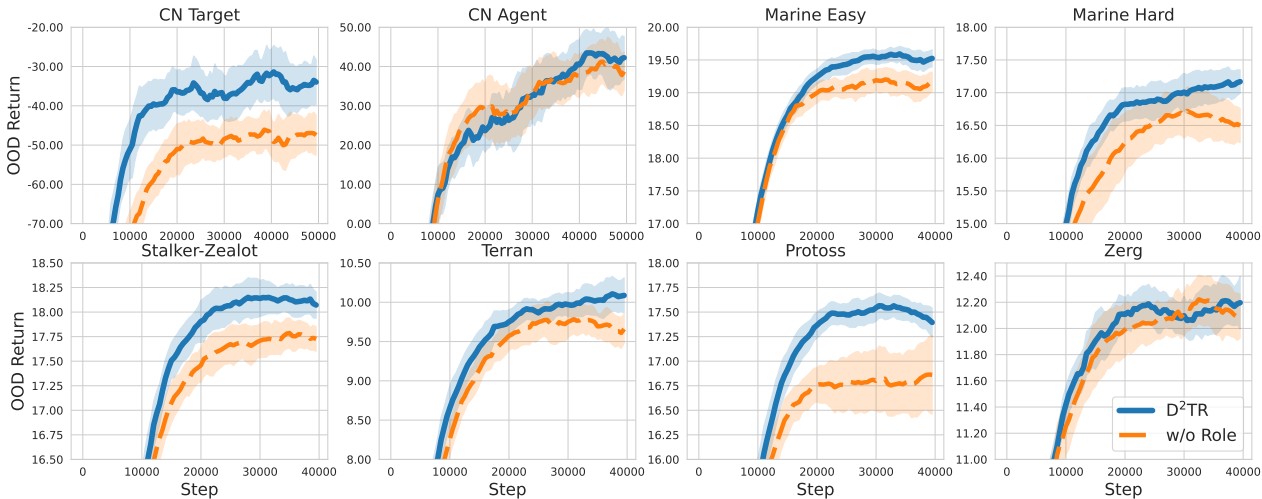

*Figure 8.* Role ablation results on more environments.

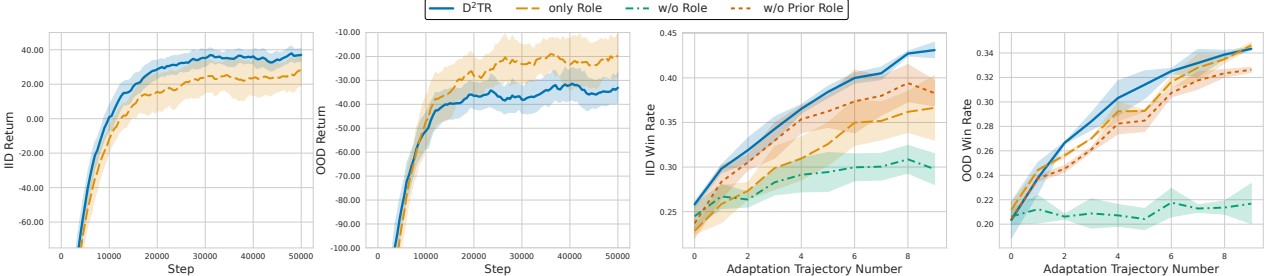

*Figure 9.* More role ablation results in CN Target.

### E.3. More Ablation Results

First, we present the results of the role ablation experiments conducted on additional environments, as shown in Figure 8. These results are consistent with those reported in the main text.

We further analyze the impact of role representations. First, we investigate the performance when only role encodings are provided as inputs to the policy, without incorporating local task encodings (**only Role**). The results are shown on the left of Figure 9. We observe that using only role encodings leads to a substantial decrease in returns under IID settings, while noticeably improving performance under OOD settings. This suggests that role encodings play a critical role in OOD generalization, whereas task encodings may introduce certain biases that hinder generalization. Nevertheless, task encodings remain crucial for accurate task identification in IID scenarios. Furthermore, as shown on the right of Figure 9, we evaluate these variants under online adaptation. In this setting, the absence of task encodings results in degraded performance for both IID and OOD tasks, indicating that task encodings, due to their relatively low reliance on the quality of the context itself, are highly beneficial during the exploration phase. Based on these observations, we conclude that task encodings remain a necessary component of the overall framework. We additionally evaluate online adaptation performance without the prior role encoder (**w/o Prior Role**, as described in Appendix C) and without role encodings (**w/o Role**), with results also shown on the right of Figure 9. When the prior role encoder is removed, performance decreases in both IID and OOD settings, validating the effectiveness of using a prior role encoder to facilitate exploration during the first $H$ steps. Finally, removing role encodings altogether leads to a significant performance drop in both settings, further demonstrating the strong generalization capability provided by role encodings.

### E.4. Model Size Analysis

Since a Transformer is used as part of the policy network, we evaluate in this section the effect of Transformer model size on policy performance without using the context encodings (Backbone). In this ablation study, Backbone Large employs

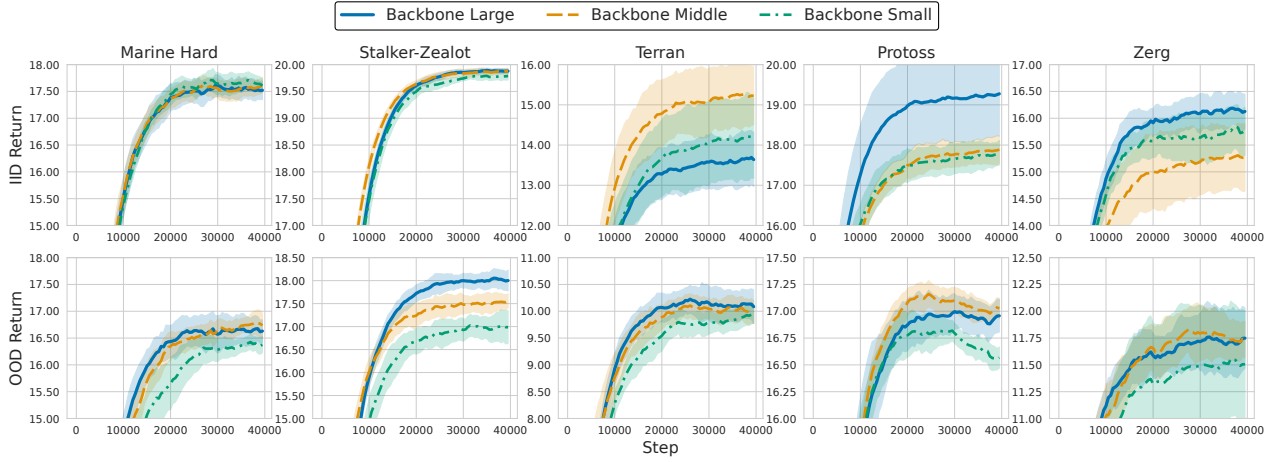

*Figure 10.* Analysis on different model sizes of Backbone.

a Transformer with 8 attention heads and a depth of 3 (consistent with the main experiments), Backbone Middle uses 3 attention heads and a depth of 2, and Backbone Small adopts 1 attention head and a depth of 1. The experimental results are shown in Figure 10. We observe that under IID settings, different model sizes do not exhibit a consistent impact on performance. In contrast, under OOD settings, Backbone Large and Backbone Middle consistently outperform Backbone Small by a clear margin. This indicates that increasing model capacity can improve policy generalization, thereby validating the scalability of $D^2TR$.

### E.5. Detailed Results for Each Task

In this section, we report the detailed results for each task across different environments in the main experiments.

*Table 5.* Detailed results in CN Target.

|  | Task | Ours | Backbone | HiSSD | ODIS | UPDET | FOCAL | CSRO | UNICORN | MG2L |
|---|---|---|---|---|---|---|---|---|---|---|
| | 0 | 0.80±0.14 | 0.28±0.05 | 0.05±0.01 | 0.01±0.01 | 0.05±0.04 | 0.79±0.04 | 0.14±0.04 | 0.36±0.07 | 0.32±0.08 |
| | 3 | **0.86±0.07** | 0.21±0.06 | 0.07±0.03 | 0.02±0.03 | 0.05±0.04 | 0.22±0.02 | 0.21±0.01 | 0.75±0.05 | 0.19±0.05 |
| | 5 | **0.88±0.00** | 0.27±0.04 | 0.07±0.07 | 0.02±0.01 | 0.03±0.02 | 0.78±0.08 | 0.22±0.00 | 0.63±0.00 | 0.32±0.10 |
| | 7 | 0.58±0.24 | 0.28±0.05 | 0.03±0.00 | 0.01±0.01 | 0.02±0.01 | 0.60±0.18 | 0.10±0.05 | **0.74±0.09** | 0.17±0.01 |
| | 10 | **0.60±0.05** | 0.24±0.06 | 0.03±0.00 | 0.03±0.02 | 0.05±0.04 | 0.38±0.00 | 0.10±0.03 | 0.52±0.06 | 0.33±0.09 |
| IID | 11 | **0.63±0.00** | 0.30±0.09 | 0.05±0.01 | 0.03±0.02 | 0.06±0.00 | 0.30±0.09 | 0.19±0.05 | 0.30±0.07 | 0.33±0.04 |
| | 13 | **0.79±0.04** | 0.39±0.11 | 0.06±0.04 | 0.03±0.04 | 0.05±0.01 | 0.74±0.06 | 0.21±0.04 | 0.57±0.13 | 0.25±0.00 |
| | 17 | **0.77±0.01** | 0.28±0.00 | 0.02±0.03 | 0.05±0.01 | 0.00±0.00 | 0.49±0.12 | 0.16±0.05 | 0.61±0.19 | 0.21±0.01 |
| | 18 | **0.82±0.10** | 0.27±0.04 | 0.02±0.01 | 0.02±0.01 | 0.03±0.00 | 0.31±0.02 | 0.19±0.02 | 0.36±0.16 | 0.22±0.02 |
| | 19 | **0.70±0.09** | 0.27±0.01 | 0.05±0.01 | 0.00±0.00 | 0.02±0.01 | 0.59±0.20 | 0.19±0.00 | 0.44±0.02 | 0.25±0.05 |
| | Mean | **0.74±0.01** | 0.28±0.04 | 0.05±0.01 | 0.02±0.00 | 0.03±0.02 | 0.52±0.00 | 0.17±0.03 | 0.53±0.09 | 0.26±0.04 |
| | 1 | **0.40±0.19** | 0.32±0.08 | 0.07±0.04 | 0.01±0.01 | 0.07±0.05 | 0.17±0.04 | 0.22±0.02 | 0.24±0.01 | 0.32±0.05 |
| | 2 | **0.54±0.08** | 0.27±0.01 | 0.07±0.01 | 0.03±0.04 | 0.08±0.07 | 0.31±0.02 | 0.19±0.05 | 0.24±0.06 | 0.22±0.07 |
| | 4 | **0.30±0.01** | 0.25±0.05 | 0.03±0.02 | 0.00±0.00 | 0.02±0.01 | 0.24±0.01 | 0.16±0.05 | 0.09±0.00 | 0.14±0.07 |
| | 6 | **0.45±0.01** | 0.24±0.04 | 0.05±0.04 | 0.02±0.03 | 0.03±0.00 | 0.21±0.06 | 0.18±0.08 | 0.16±0.05 | 0.27±0.01 |
| | 8 | 0.33±0.07 | 0.29±0.08 | 0.03±0.02 | 0.00±0.00 | 0.02±0.01 | **0.36±0.11** | 0.21±0.04 | 0.22±0.05 | 0.27±0.14 |
| OOD | 9 | **0.50±0.05** | 0.31±0.02 | 0.07±0.06 | 0.00±0.00 | 0.03±0.02 | 0.24±0.01 | 0.15±0.01 | 0.17±0.01 | 0.21±0.01 |
| | 12 | 0.17±0.04 | 0.22±0.02 | 0.05±0.03 | 0.04±0.01 | 0.05±0.04 | **0.27±0.06** | 0.17±0.07 | 0.24±0.01 | 0.24±0.04 |
| | 14 | **0.47±0.13** | 0.26±0.10 | 0.08±0.04 | 0.03±0.00 | 0.03±0.02 | 0.27±0.06 | 0.14±0.04 | 0.21±0.01 | 0.28±0.05 |
| | 15 | 0.20±0.11 | **0.27±0.04** | 0.07±0.01 | 0.01±0.01 | 0.08±0.04 | 0.26±0.10 | 0.13±0.00 | 0.22±0.02 | 0.24±0.01 |
| | 16 | **0.55±0.22** | 0.36±0.04 | 0.03±0.02 | 0.00±0.00 | 0.03±0.00 | 0.27±0.06 | 0.22±0.02 | 0.21±0.04 | 0.22±0.05 |
| | Mean | **0.39±0.03** | 0.28±0.03 | 0.06±0.01 | 0.02±0.01 | 0.05±0.03 | 0.26±0.00 | 0.17±0.04 | 0.20±0.02 | 0.24±0.02 |

*Table 6.* Detailed results in CN Agent.

| Task | | Ours | Backbone | HiSSD | ODIS | UPDET | FOCAL | CSRO | UNICORN | MG2L |
|---|---|---|---|---|---|---|---|---|---|---|
| IID | 2 Agent | **1.00±0.00** | **1.00±0.00** | **1.00±0.00** | 0.75±0.16 | 0.77±0.09 | 0.99±0.01 | **1.00±0.00** | **1.00±0.00** | **1.00±0.00** |
| | 4 Agent | 0.77±0.09 | 0.78±0.00 | 0.47±0.10 | 0.44±0.12 | 0.41±0.02 | 0.77±0.04 | 0.72±0.13 | 0.79±0.04 | **0.83±0.03** |
| | Mean | 0.89±0.04 | 0.89±0.00 | 0.74±0.05 | 0.59±0.13 | 0.69±0.06 | 0.88±0.01 | 0.86±0.07 | 0.90±0.02 | **0.92±0.01** |
| OOD | 3 Agent | **0.93±0.01** | 0.88±0.00 | 0.77±0.06 | 0.45±0.19 | 0.74±0.01 | 0.88±0.00 | 0.84±0.08 | 0.83±0.01 | 0.91±0.02 |
| | 5 Agent | **0.66±0.02** | **0.66±0.02** | 0.33±0.01 | 0.21±0.15 | 0.31±0.02 | 0.54±0.04 | 0.48±0.07 | 0.54±0.01 | 0.53±0.09 |
| | Mean | **0.79±0.02** | 0.77±0.01 | 0.54±0.04 | 0.33±0.17 | 0.53±0.00 | 0.71±0.02 | 0.66±0.07 | 0.69±0.00 | 0.72±0.06 |

*Table 7.* Detailed results in Marine Easy.

| Task | | Ours | Backbone | HiSSD | ODIS | UPDET | FOCAL | CSRO | UNICORN | MG2L |
|---|---|---|---|---|---|---|---|---|---|---|
| IID | 3m | **1.00±0.00** | 0.99±0.01 | 0.99±0.01 | 0.98±0.03 | 0.60±0.08 | 0.99±0.01 | 0.88±0.08 | 0.97±0.02 | 0.97±0.00 |
| | 5m | **1.00±0.00** | **1.00±0.00** | **1.00±0.00** | 0.80±0.29 | 0.47±0.10 | **1.00±0.00** | 0.90±0.08 | **1.00±0.00** | **1.00±0.00** |
| | 7m | **1.00±0.00** | **1.00±0.00** | **1.00±0.00** | 0.98±0.03 | 0.86±0.09 | 0.99±0.01 | **1.00±0.00** | **1.00±0.00** | **1.00±0.00** |
| | 10m | **1.00±0.00** | **1.00±0.00** | 0.99±0.01 | 0.51±0.13 | 0.38±0.15 | 0.99±0.01 | 0.88±0.10 | **1.00±0.00** | 0.99±0.01 |
| | Mean | **1.00±0.00** | **1.00±0.00** | 0.99±0.01 | 0.82±0.09 | 0.57±0.02 | 0.99±0.00 | 0.92±0.06 | 0.99±0.01 | 0.99±0.00 |
| OOD | 4m | 0.85±0.06 | 0.96±0.01 | 0.89±0.07 | 0.75±0.22 | 0.49±0.12 | 0.86±0.09 | 0.44±0.28 | 0.93±0.01 | **0.99±0.01** |
| | 6m | **1.00±0.00** | **1.00±0.00** | 0.99±0.01 | 0.77±0.14 | 0.97±0.00 | 0.78±0.16 | 0.91±0.00 | **1.00±0.00** | **1.00±0.00** |
| | 8m | **0.97±0.00** | 0.94±0.05 | 0.96±0.01 | 0.58±0.26 | 0.32±0.08 | 0.81±0.02 | 0.78±0.16 | **0.97±0.02** | 0.92±0.07 |
| | 9m | 0.99±0.01 | 0.99±0.01 | **1.00±0.00** | 0.38±0.30 | 0.38±0.31 | 0.97±0.00 | 0.77±0.11 | 0.99±0.01 | 0.99±0.01 |
| | 11m | **0.97±0.04** | **0.97±0.00** | 0.94±0.05 | 0.59±0.30 | 0.27±0.19 | **0.97±0.00** | 0.52±0.27 | **0.97±0.00** | 0.93±0.04 |
| | 12m | 0.93±0.06 | 0.67±0.11 | **0.94±0.02** | 0.26±0.13 | 0.03±0.02 | 0.86±0.07 | 0.19±0.00 | 0.72±0.05 | 0.69±0.10 |
| | Mean | **0.95±0.01** | 0.92±0.01 | **0.95±0.00** | 0.56±0.14 | 0.41±0.08 | 0.88±0.04 | 0.60±0.05 | 0.93±0.01 | 0.92±0.03 |

*Table 8.* Detailed results in Marine Hard.

| Task | | Ours | Backbone | HiSSD | ODIS | UPDET | FOCAL | CSRO | UNICORN | MG2L |
|---|---|---|---|---|---|---|---|---|---|---|
| IID | 3m | **1.00±0.00** | 1.00±0.00 | **1.00±0.00** | 1.00±0.00 | 0.53±0.20 | **1.00±0.00** | 0.11±0.09 | **1.00±0.00** | 0.99±0.01 |
| | 5m_vs_6m | 0.45±0.08 | 0.40±0.01 | **0.54±0.24** | 0.13±0.08 | 0.02±0.01 | 0.32±0.22 | 0.00±0.00 | 0.42±0.07 | 0.43±0.01 |
| | 9m_vs_10m | 0.91±0.05 | 0.94±0.05 | **0.97±0.02** | 0.44±0.33 | 0.11±0.04 | 0.52±0.37 | 0.00±0.00 | 0.85±0.08 | 0.96±0.01 |
| | 10m_vs_12m | 0.40±0.08 | 0.42±0.07 | **0.55±0.01** | 0.27±0.01 | 0.00±0.00 | 0.27±0.19 | 0.00±0.00 | 0.21±0.17 | 0.36±0.02 |
| | Mean | 0.69±0.05 | 0.69±0.01 | **0.77±0.06** | 0.46±0.10 | 0.17±0.04 | 0.53±0.20 | 0.03±0.02 | 0.62±0.04 | 0.68±0.02 |
| OOD | 4m | 0.91±0.05 | 0.91±0.02 | **0.99±0.01** | 0.39±0.27 | 0.57±0.08 | 0.64±0.16 | 0.39±0.32 | 0.86±0.04 | 0.97±0.02 |
| | 5m | 0.96±0.04 | 0.90±0.08 | **1.00±0.00** | 0.41±0.08 | 0.93±0.01 | 0.88±0.08 | 0.53±0.05 | 0.99±0.01 | 0.99±0.01 |
| | 7m_vs_8m | 0.38±0.09 | 0.33±0.04 | **0.61±0.09** | 0.16±0.10 | 0.03±0.00 | 0.11±0.09 | 0.05±0.01 | 0.28±0.18 | 0.41±0.00 |
| | 8m_vs_9m | **0.59±0.11** | 0.31±0.02 | 0.57±0.26 | 0.16±0.02 | 0.03±0.00 | 0.38±0.31 | 0.02±0.01 | 0.24±0.09 | 0.47±0.13 |
| | 10m_vs_11m | 0.98±0.01 | 0.82±0.13 | **0.99±0.01** | 0.38±0.08 | 0.11±0.02 | 0.71±0.24 | 0.03±0.02 | 0.49±0.40 | 0.97±0.02 |
| | 13m_vs_15m | **0.16±0.09** | 0.05±0.01 | **0.16±0.08** | 0.13±0.10 | 0.02±0.01 | 0.00±0.00 | 0.00±0.00 | 0.02±0.01 | 0.02±0.01 |
| | Mean | 0.66±0.02 | 0.55±0.02 | **0.72±0.07** | 0.27±0.05 | 0.28±0.02 | 0.45±0.15 | 0.17±0.06 | 0.47±0.11 | 0.63±0.02 |

*Table 9.* Detailed results in Stalker-Zealot.

| Task | | Ours | Backbone | HiSSD | ODIS | UPDET | FOCAL | CSRO | UNICORN | MG2L |
|---|---|---|---|---|---|---|---|---|---|---|
| IID | 2s3z | 0.99±0.01 | 0.97±0.00 | 0.97±0.00 | 0.97±0.02 | 0.07±0.05 | 0.35±0.03 | 0.94±0.05 | 0.94±0.02 | **1.00±0.00** |
| | 2s4z | **1.00±0.00** | 0.97±0.00 | 0.89±0.07 | 0.75±0.07 | 0.13±0.03 | 0.05±0.01 | 0.99±0.01 | 0.97±0.02 | 0.99±0.01 |
| | 3s5z | 0.97±0.00 | **1.00±0.00** | 0.91±0.02 | 0.85±0.03 | 0.03±0.00 | 0.02±0.01 | 0.94±0.05 | 0.94±0.00 | 0.99±0.01 |
| | Mean | **0.99±0.00** | 0.98±0.00 | 0.93±0.01 | 0.86±0.00 | 0.07±0.02 | 0.14±0.02 | 0.96±0.00 | 0.95±0.00 | **0.99±0.00** |
| OOD | 1s3z | 0.74±0.04 | 0.58±0.01 | **0.88±0.10** | 0.25±0.16 | 0.05±0.04 | 0.14±0.07 | 0.49±0.09 | 0.72±0.02 | 0.66±0.05 |
| | 1s4z | **0.82±0.05** | 0.80±0.11 | 0.60±0.05 | 0.66±0.02 | 0.10±0.03 | 0.07±0.05 | 0.63±0.05 | 0.74±0.04 | 0.81±0.00 |
| | 1s5z | **0.74±0.01** | 0.68±0.01 | 0.54±0.08 | 0.20±0.11 | 0.16±0±.08 | 0.02±0.01 | 0.49±0.01 | 0.69±0.02 | 0.66±0.00 |
| | 2s5z | 0.86±0.02 | **0.96±0.01** | 0.56±0.00 | 0.58±0.11 | 0.05±0.04 | 0.00±0.00 | 0.85±0.03 | 0.93±0.01 | 0.83±0.04 |
| | 3s3z | 0.81±0.02 | 0.74±0.04 | 0.77±0.04 | 0.67±0.07 | 0.05±0.01 | 0.17±0.11 | 0.54±0.08 | 0.66±0.02 | **0.82±0.01** |
| | 3s4z | **1.00±0.00** | 0.94±0.00 | 0.88±0.03 | 0.89±0.04 | 0.10±0.03 | 0.07±0.05 | 0.83±0.04 | 0.88±0.05 | 0.94±0.02 |
| | 4s3z | 0.53±0.05 | 0.38±0.08 | 0.36±0.09 | 0.22±0.02 | 0.00±0.00 | 0.17±0.09 | 0.41±0.08 | 0.49±0.17 | **0.58±0.07** |
| | 4s4z | 0.44±0.08 | 0.42±0.07 | 0.13±0.10 | **0.47±0.18** | 0.05±0.04 | 0.02±0.01 | 0.36±0.04 | 0.32±0.10 | **0.47±0.07** |
| | 4s5z | 0.51±0.10 | 0.33±0.04 | 0.30±0.14 | **0.63±0.10** | 0.02±0.01 | 0.00±0.00 | 0.38±0.15 | 0.30±0.04 | 0.28±0.02 |
| | Mean | **0.72±0.00** | 0.65±0.02 | 0.56±0.04 | 0.51±0.04 | 0.06±0.02 | 0.07±0.04 | 0.55±0.01 | 0.64±0.03 | 0.67±0.01 |

*Table 10.* Detailed results in Terran.

| Task | Ours | Backbone | HiSSD | ODIS | UPDET | FOCAL | CSRO | UNICORN | MG2L |
|---|---|---|---|---|---|---|---|---|---|
| IID | | | | | | | | | |
| MarineMarauder_5v5 | 0.63±0.05 | 0.54±0.04 | 0.61±0.04 | 0.34±0.07 | 0.09±0.00 | 0.72±0.05 | 0.61±0.02 | **0.74±0.01** | 0.68±0.04 |
| MarineMedivac_5v5 | **0.51±0.05** | 0.39±0.04 | 0.40±0.01 | 0.23±0.06 | 0.08±0.07 | 0.47±0.02 | 0.47±0.07 | 0.34±0.00 | 0.47±0.00 |
| MarauderMedivac_10v10 | 0.23±0.07 | 0.14±0.04 | 0.30±0.07 | 0.04±0.04 | 0.00±0.00 | 0.19±0.00 | 0.21±0.01 | **0.36±0.07** | 0.22±0.02 |
| MarineMarauder_10v11 | 0.53±0.08 | 0.60±0.08 | 0.44±0.02 | 0.08±0.01 | 0.00±0.00 | 0.52±0.04 | 0.61±0.02 | **0.65±0.01** | 0.45±0.11 |
| MarineMedivac_10v11 | 0.69±0.05 | 0.60±0.03 | 0.41±0.02 | 0.15±0.11 | 0.09±0.00 | **0.71±0.04** | 0.69±0.08 | 0.58±0.04 | 0.69±0.05 |
| Mean | 0.52±0.01 | 0.46±0.00 | 0.43±0.03 | 0.17±0.04 | 0.06±0.01 | 0.52±0.01 | 0.52±0.04 | **0.54±0.02** | 0.50±0.02 |
| OOD | | | | | | | | | |
| 1m3m1m | 0.20±0.06 | 0.21±0.01 | **0.32±0.05** | 0.08±0.04 | 0.03±0.02 | 0.11±0.02 | 0.21±0.04 | 0.25±0.02 | 0.19±0.05 |
| 2m2m1m | 0.24±0.06 | 0.17±0.01 | 0.22±0.00 | 0.10±0.04 | 0.05±0.04 | 0.24±0.06 | **0.33±0.09** | 0.29±0.10 | 0.32±0.10 |
| 3m1m1m | **0.53±0.04** | 0.53±0.02 | 0.49±0.01 | 0.22±0.12 | 0.03±0.02 | 0.49±0.06 | 0.42±0.09 | **0.53±0.04** | **0.53±0.13** |
| 3m3m4m | 0.39±0.10 | 0.33±0.04 | 0.28±0.02 | 0.09±0.05 | 0.02±0.01 | 0.33±0.01 | **0.43±0.01** | 0.33±0.01 | 0.28±0.00 |
| 1m5m4m | **0.32±0.11** | 0.13±0.03 | 0.13±0.00 | 0.09±0.09 | 0.00±0.00 | 0.21±0.03 | 0.27±0.04 | 0.28±0.02 | 0.19±0.02 |
| 5m1m4m | **0.68±0.04** | 0.66±0.04 | 0.61±0.07 | 0.24±0.10 | 0.14±0.09 | 0.57±0.05 | 0.53±0.02 | 0.44±0.08 | 0.66±0.07 |
| 5m3m2m_vs_6m3m2m | 0.16±0.07 | 0.10±0.03 | **0.29±0.08** | 0.03±0.02 | 0.00±0.00 | 0.28±0.02 | 0.27±0.07 | **0.29±0.13** | 0.21±0.01 |
| 2m6m2m_vs_3m6m2m | 0.04±0.03 | 0.02±0.01 | 0.11±0.02 | 0.04±0.01 | 0.00±0.00 | 0.11±0.02 | 0.03±0.00 | **0.14±0.07** | 0.00±0.00 |
| 4m3m3m_vs_5m3m3m | 0.11±0.06 | 0.14±0.04 | **0.22±0.10** | 0.07±0.07 | 0.00±0.00 | 0.17±0.04 | 0.17±0.01 | 0.10±0.05 | 0.06±0.02 |
| 8m2m10m | 0.04±0.04 | 0.16±0.13 | **0.22±0.10** | 0.03±0.00 | 0.03±0.02 | 0.00±0.00 | 0.00±0.00 | 0.02±0.01 | 0.00±0.00 |
| 6m9m5m | 0.02±0.01 | 0.02±0.01 | **0.10±0.03** | 0.03±0.02 | 0.00±0.00 | 0.00±0.00 | 0.00±0.00 | 0.00±0.00 | 0.00±0.00 |
| 5m8m7m | 0.01±0.01 | 0.00±0.00 | **0.09±0.00** | 0.01±0.01 | 0.00±0.00 | 0.00±0.00 | 0.00±0.00 | 0.00±0.00 | 0.00±0.00 |
| Mean | 0.23±0.01 | 0.20±0.02 | **0.26±0.02** | 0.09±0.03 | 0.03±0.02 | 0.21±0.01 | 0.22±0.01 | 0.22±0.01 | 0.20±0.01 |

*Table 11.* Detailed results in Protoss.

| | Task | Ours | Backbone | HiSSD | ODIS | UPDET | FOCAL | CSRO | UNICORN | MG2L |
|---|---|---|---|---|---|---|---|---|---|---|
| IID | StalkerZealot_5v5 | 0.58±0.04 | **0.75±0.02** | 0.74±0.06 | 0.24±0.18 | 0.08±0.01 | 0.71±0.01 | 0.67±0.07 | 0.72±0.07 | 0.63±0.03 |
| | StalkerColossus_5v5 | **0.82±0.08** | 0.77±0.04 | 0.69±0.00 | 0.49±0.06 | 0.09±0.00 | 0.72±0.02 | 0.69±0.13 | 0.78±0.05 | 0.81±0.00 |
| | ZealotColossus_10v10 | 0.85±0.03 | 0.86±0.04 | 0.77±0.04 | 0.37±0.11 | 0.11±0.07 | 0.82±0.01 | 0.72±0.00 | **0.97±0.02** | 0.90±0.02 |
| | StalkerZealot_10v11 | **0.35±0.10** | 0.16±0.03 | 0.16±0.02 | 0.05±0.06 | 0.00±0.00 | 0.28±0.02 | 0.22±0.02 | 0.24±0.06 | 0.25±0.02 |
| | StalkerColossus_10v11 | 0.57±0.05 | **0.74±0.01** | 0.47±0.00 | 0.17±0.01 | 0.05±0.04 | 0.49±0.04 | 0.65±0.01 | 0.52±0.06 | 0.72±0.05 |
| | Mean | 0.63±0.04 | **0.66±0.02** | 0.56±0.02 | 0.27±0.06 | 0.07±0.02 | 0.60±0.01 | 0.59±0.02 | 0.65±0.01 | **0.66±0.00** |
| OOD | 1s3z1c | 0.71±0.01 | 0.69±0.05 | 0.69±0.02 | 0.40±0.14 | 0.11±0.02 | **0.75±0.02** | 0.66±0.08 | **0.75±0.07** | 0.69±0.02 |
| | 2s2z1c | **0.75±0.02** | 0.72±0.00 | 0.56±0.00 | 0.35±0.11 | 0.06±0.02 | 0.64±0.07 | 0.57±0.10 | 0.70±0.09 | 0.74±0.06 |
| | 1s1z3c | 0.82±0.01 | 0.75±0.02 | **0.85±0.03** | 0.53±0.10 | 0.11±0.02 | 0.81±0.02 | 0.78±0.00 | 0.82±0.01 | 0.78±0.02 |
| | 4s3z3c | **0.85±0.03** | 0.75±0.03 | 0.64±0.04 | 0.37±0.08 | 0.16±0.02 | 0.64±0.07 | 0.69±0.02 | 0.63±0.03 | 0.75±0.02 |
| | 5s2z3c | 0.75±0.07 | 0.75±0.03 | 0.68±0.01 | 0.40±0.06 | 0.08±0.01 | 0.76±0.10 | 0.69±0.02 | 0.74±0.04 | **0.86±0.09** |
| | 2s5z3c | **0.85±0.05** | 0.81±0.00 | 0.63±0.03 | 0.33±0.10 | 0.05±0.01 | 0.74±0.04 | 0.77±0.06 | 0.83±0.05 | 0.80±0.01 |
| | 3s2z5c_vs_3s3z5c | **0.82±0.01** | 0.80±0.07 | 0.59±0.00 | 0.28±0.08 | 0.08±0.01 | 0.78±0.00 | 0.75±0.05 | 0.70±0.11 | 0.64±0.07 |
| | 2s7z1c_vs_2s7z2c | **0.15±0.01** | 0.11±0.02 | 0.11±0.02 | 0.03±0.02 | 0.00±0.00 | 0.10±0.03 | 0.08±0.01 | 0.08±0.01 | 0.13±0.03 |
| | 5s4z1c_vs_6s4z1c | 0.30±0.01 | 0.36±0.02 | 0.36±0.04 | 0.03±0.02 | 0.00±0.00 | 0.31±0.00 | 0.33±0.04 | 0.30±0.01 | **0.41±0.10** |
| | 5s6z9c | 0.16±0.08 | 0.17±0.04 | **0.36±0.07** | 0.03±0.02 | 0.03±0.02 | 0.16±0.10 | 0.22±0.05 | 0.17±0.07 | 0.11±0.09 |
| | 5s8z7c | 0.22±0.04 | 0.14±0.07 | **0.49±0.14** | 0.00±0.00 | 0.03±0.02 | 0.14±0.09 | 0.17±0.04 | 0.19±0.05 | 0.07±0.05 |
| | 8s7z5c | 0.06±0.02 | 0.11±0.07 | **0.39±0.14** | 0.01±0.01 | 0.02±0.01 | 0.11±0.07 | 0.06±0.02 | 0.06±0.02 | 0.03±0.02 |
| | Mean | **0.54±0.02** | 0.52±0.02 | 0.53±0.03 | 0.23±0.05 | 0.06±0.01 | 0.49±0.04 | 0.48±0.02 | 0.50±0.02 | 0.49±0.02 |

*Table 12.* Detailed results in Zerg.

| | Task | Ours | Backbone | HiSSD | ODIS | UPDET | FOCAL | CSRO | UNICORN | MG2L |
|---|---|---|---|---|---|---|---|---|---|---|
| IID | ZerglingHydralisk_5v5 | 0.43±0.01 | 0.40±0.01 | 0.38±0.03 | 0.19±0.06 | 0.10±0.05 | 0.36±0.07 | 0.44±0.02 | 0.45±0.04 | **0.53±0.02** |
| | ZerglingBaneling_5v5 | 0.78±0.05 | 0.83±0.04 | 0.83±0.07 | 0.83±0.04 | 0.66±0.00 | **0.85±0.03** | 0.82±0.01 | 0.77±0.01 | 0.77±0.01 |
| | HydraliskBaneling_10v10 | 0.29±0.03 | **0.43±0.01** | 0.17±0.01 | 0.08±0.01 | 0.06±0.02 | 0.18±0.08 | 0.28±0.00 | 0.40±0.01 | 0.42±0.11 |
| | ZerglingHydralisk_10v11 | **0.40±0.04** | 0.28±0.00 | 0.31±0.02 | 0.08±0.08 | 0.00±0.00 | **0.40±0.13** | 0.38±0.00 | 0.34±0.00 | 0.33±0.04 |
| | ZerglingBaneling_10v11 | 0.79±0.09 | 0.81±0.02 | 0.81±0.00 | 0.70±0.10 | 0.63±0.05 | 0.85±0.05 | **0.88±0.05** | 0.82±0.01 | 0.72±0.02 |
| | Mean | 0.54±0.02 | 0.55±0.00 | 0.50±0.01 | 0.38±0.04 | 0.29±0.00 | 0.53±0.04 | **0.56±0.01** | **0.56±0.00** | 0.55±0.03 |
| OOD | 2z1h2b | 0.15±0.08 | 0.15±0.01 | **0.16±0.02** | 0.15±0.05 | 0.06±0.00 | 0.08±0.01 | 0.08±0.01 | 0.09±0.00 | 0.05±0.01 |
| | 1z3h1b | 0.27±0.03 | 0.25±0.02 | 0.11±0.02 | 0.09±0.03 | 0.00±0.00 | 0.22±0.05 | 0.22±0.00 | 0.19±0.00 | **0.32±0.05** |
| | 3z1h1b | 0.13±0.13 | 0.16±0.00 | 0.25±0.05 | 0.09±0.05 | 0.08±0.04 | 0.19±0.00 | 0.21±0.01 | **0.35±0.03** | 0.27±0.01 |
| | 3z5h2b | 0.38±0.10 | 0.33±0.07 | 0.25±0.07 | 0.12±0.07 | 0.02±0.01 | 0.35±0.08 | 0.33±0.01 | 0.22±0.05 | **0.43±0.01** |
| | 4z3h3b | 0.40±0.03 | 0.30±0.01 | 0.22±0.07 | 0.05±0.01 | 0.00±0.00 | 0.33±0.01 | 0.28±0.02 | 0.31±0.00 | **0.44±0.08** |
| | 4z1h5b | **0.52±0.08** | 0.33±0.11 | 0.25±0.07 | 0.07±0.05 | 0.11±0.04 | 0.45±0.01 | 0.43±0.01 | 0.51±0.07 | 0.33±0.09 |
| | 4z2h4b_vs_5z2h4b | 0.19±0.02 | 0.05±0.04 | 0.11±0.02 | 0.01±0.01 | 0.02±0.01 | 0.05±0.04 | 0.10±0.05 | **0.22±0.02** | 0.08±0.01 |
| | 2z4h4b_vs_2z5h4b | 0.02±0.01 | 0.09±0.00 | 0.08±0.01 | 0.01±0.01 | 0.00±0.00 | 0.02±0.01 | 0.06±0.02 | **0.17±0.01** | 0.05±0.01 |
| | 3z4h3b_vs_3z4h4b | **0.27±0.03** | 0.17±0.01 | 0.16±0.05 | 0.01±0.01 | 0.02±0.01 | 0.10±0.03 | 0.10±0.05 | 0.13±0.03 | 0.05±0.04 |
| | 10z3h7b | 0.00±0.00 | 0.03±0.02 | **0.11±0.07** | 0.02±0.01 | 0.02±0.01 | 0.02±0.01 | 0.02±0.01 | 0.02±0.01 | 0.00±0.00 |
| | 7z11h2b | 0.00±0.00 | 0.00±0.00 | **0.10±0.03** | 0.00±0.00 | 0.03±0.02 | 0.00±0.00 | 0.00±0.00 | 0.00±0.00 | 0.03±0.02 |
| | 5z6h9b | **0.02±0.03** | 0.00±0.00 | **0.02±0.01** | 0.01±0.01 | **0.02±0.01** | **0.02±0.03** | 0.00±0.00 | 0.00±0.00 | **0.02±0.01** |
| | Mean | **0.20±0.02** | 0.16±0.01 | 0.15±0.00 | 0.05±0.01 | 0.03±0.01 | 0.15±0.00 | 0.15±0.01 | 0.18±0.00 | 0.17±0.01 |

