# OpenReview forum: "Decentralized and Disentangled Task–Role Representation Learning for Generalizable Offline Multi-Agent Meta Reinforcement Learning"
_ICML.cc/2026/Conference — ICML 2026 regular_

### Official Review · Reviewer_WZaw · 2026-03-11

**Soundness:** 3
**Presentation:** 3
**Significance:** 3
**Originality:** 3
**Overall Recommendation:** 4
**Confidence:** 4

**Summary:**

This paper studies offline multi-agent meta-reinforcement learning, where agents aim to learn decentralized policies that maximize the expected return across multiple tasks using offline data. The paper proposes a new task-focused representation learning framework based on mutual information objectives that incorporate both global and local embeddings of offline trajectories. In addition, the paper introduces the use of LLMs to identify agents’ individual roles, enabling the incorporation of role embeddings into subsequent policy learning. Experiments on multiple benchmarks, including SMAC tasks, show that policies learned using the proposed representation learning approach outperform strong baselines, particularly under out-of-distribution (OOD) settings.

**Compliance With Llm Reviewing Policy:**

Affirmed.

**Key Questions For Authors:**

1. Could the authors address the concerns raised above regarding the dependence on LLMs for role identification and the practical procedure for separating behavior-related and task-related context?

2. As described in the paper, the encoders take either an entire trajectory (local level) or a set of trajectories (global level) as input, and the resulting embeddings are used for Q-value, V-value, and policy learning. However, during policy execution, the full trajectory information will not be available in advance. How is the learned policy executed at test time under this constraint?

**Limitations:**

Yes.

**Strengths And Weaknesses:**

*** Strengths
1. The proposed framework for learning both local and global task embeddings integrates multiple objectives that capture different aspects of task representation. These include contrastive learning objectives for task embeddings at both the local and global levels, alignment between local and global embeddings within each task to enforce cooperative consensus among agents, and a reconstruction objective that models the relationship between agent behavior and task-related context.

2. The paper demonstrates an interesting application of LLMs to infer the individual roles of agents in each task when such information is not available in advance. The resulting role identification enables the integration of role embeddings into policy learning, which appears to improve policy quality.

3. The paper presents extensive experiments across multiple benchmarks including SMAC and shows that the proposed representation learning approach consistently outperforms several strong baselines.


*** Weaknesses:
1. The effectiveness of role identification appears to depend heavily on the capability of the chosen LLM and the semantic interpretability of the environment. It is unclear how sensitive the approach is to the choice of LLM or how the method would perform in environments with limited or ambiguous semantic information.

2.The paper separates observations into behavior-related and task-related contexts (denoted as (o, o'). However, it is unclear how such a decomposition is performed in practice. For example, beyond rewards and actions, the observation space is divided into behavior and task-related components, but the methodology for performing this separation is not clearly described.

---

> ### Author Rebuttal · Authors · 2026-03-30
>
> Thank you for your kind and careful reviews. We appreciate your feedback and hope that our response addresses your concerns. Additional experiments are provided in [https://telling-floor-898.notion.site/Additional-experimental-results-for-ICML-paper-4780-333c2fed721a803d9aa3fea9cbf2ccaf](https://www.notion.so/Additional-experimental-results-for-ICML-paper-4780-333c2fed721a803d9aa3fea9cbf2ccaf?pvs=21)
>
> ### Q1 More analysis on LLM-based role annotation
>
> Thank you for the valuable suggestion. We have added a sensitivity study on **LLM-based role annotation** in **Table 1 of the experiment link**.
>
> First, comparing our method with variants using **DeepSeek V3.2** and **GPT-5-mini** shows that, in both the **code-based role** setting on **CN Target** and the **description-clustering-based** setting on **Stalker-Zealot**, the choice of LLM has only a limited effect on overall performance. This suggests that our method is reasonably robust to the specific LLM used.
>
> Second, comparison with a **clustering-based baseline without LLMs** reveals clear environment-dependent differences. In **CN Target**, **Clustering Role** generalizes poorly and does not outperform the **Backbone**, highlighting the need for external knowledge in role labeling. In **Stalker-Zealot**, **Clustering Role** is slightly worse than our method but still better than the Backbone, showing that role information is beneficial. However, clustering can only separate agents of different races within a task and cannot effectively merge same-race agents across tasks, limiting knowledge sharing and likely explaining its gap from our **LLM-based description** method.
>
> Finally, in **description-clustering-based Stalker-Zealot**, we add two more baselines: **New Seed**, which asks the LLM to perform a fresh round of role labeling, and **Simple Prompt**, which removes the example (*“for example, a tank that lures enemies on the front line, or a kiter dealing damage from the back line”*) in our prompt. Both achieve performance similar to the full method, suggesting that our approach is also reasonably robust to **LLM randomness** and **prompt wording**.
>
> In addition, our current method does require that trajectories in the environment be describable in clear natural language, which makes it difficult to directly extend to settings where trajectory semantics are ambiguous or hard to verbalize. For such environments, new and more effective role-labeling methods will likely be needed, and we view this as a promising direction for future work.
>
> ### Q2 Separating behavior-related and task-related context
>
> In fact, for a transition $(o,a,r,o’)$, we do not further decompose the observation $o$ or the next observation $o’$. Instead, the separation is performed at the level of the transition itself. Specifically, **the observation $o$ and action $a$ are treated as behavior-related context, while the reward $r$ and next observation $o’$ are regarded as task-related context**.
>
> ### Q3 The context trajectory during policy deployment
>
> During the policy deployment phase, only the **individual trajectory of a single agent** is required as context to infer the local task representation; access to the full global trajectory involving multiple agents is not necessary.
>
> Regarding the source of individual context, we follow standard practices in prior meta-RL literature [1-3] and consider two settings. First, the policy is **provided with individual demonstration trajectories** as context. Second, no demonstrations are given, and the policy is required to **actively collect trajectories from the environment to construct the context**. These two settings correspond to the results reported in **Tab. 1 and Fig. 2**, respectively.
>
> [1] Li L, Yang R, Luo D. Focal: Efficient fully-offline meta-reinforcement learning via distance metric learning and behavior regularization[J]. arXiv preprint arXiv:2010.01112, 2020.
>
> [2] Gao Y, Zhang R, Guo J, et al. Context shift reduction for offline meta-reinforcement learning[J]. Advances in Neural Information Processing Systems, 2023, 36: 80024-80043.
>
> [3] Li L, Zhang H, Zhang X, et al. Towards an information theoretic framework of context-based offline meta-reinforcement learning[J]. Advances in Neural Information Processing Systems, 2024, 37: 75642-75667.

---

> > ### Author Rebuttal · Reviewer_WZaw · 2026-04-03
> >
> > Thank you for your rebuttal!
> >
> > Can you please clarify more on individual demonstration trajectories at the test time? For example, at current state s, the agent will use past complete trajectories to generate the embeddings which will be used as additional conditions to determine the action to take at s?

---

> > > ### Author Response · Authors · 2026-04-03
> > >
> > > Thank you very much for your response and for your positive assessment of our rebuttal. We sincerely appreciate your feedback and hope that our response has addressed your additional questions. Of course, we can clarify more on individual demonstration trajectories at the test time.
> > >
> > > Under the demonstration setting, each policy is provided with a global trajectory corresponding to the task as a demonstration. For each agent, given its local observation $o$, it extracts the corresponding individual trajectory from the demonstration as context, and uses this context to obtain an individual embedding. The action $a$ is then **determined based on both $o$ and the individual embedding**.
> > >
> > > Under the online adaptation setting, each policy starts from **a randomly initialized individual embedding**. It then iteratively uses **a past complete trajectory** collected in the current environment as context to update the individual embedding, and selects actions $a$ conditioned on both $o$ and the updated embedding. This meta-testing process is introduced by PEARL [1] in off-policy RL, and is the most common way in context-based meta RL. The detailed online adaptation procedure could be found in Algorithm 2 in the Appendix.
> > >
> > > [1] Efficient Off-Policy Meta-Reinforcement Learning via Probabilistic Context Variables

---

### Official Review · Reviewer_xKZi · 2026-03-12

**Soundness:** 2
**Presentation:** 2
**Significance:** 1
**Originality:** 2
**Overall Recommendation:** 3
**Confidence:** 3

**Summary:**

The paper proposes a context-based meta  multi-agent RL method named D$^2$TR,  incorporating both task and role representation learning. Experiments conducted on CN, SMAC, and SMACv2 show strong generalization performance over tasks under both IID and OOD settings.

**Compliance With Llm Reviewing Policy:**

Affirmed.

**Key Questions For Authors:**

1. Minor typo: the brackets seem to be misused in eq.8 and eq.9. The formulation should be  $(\beta L_{inro}(...) + (1-\beta)L_{recon}(...)))$, rather than$(\beta L_{inro}(...) + (1-\beta))L_{recon}(...))$.


2. Could the authors compare LLM-generated role labels with alternative approaches, such as human-generated labeling or heuristic-based labeling?

3. Could the author provide experimental results on the stability and consistency of the labels generated by the LLMs?

4. In fig.4(b), the decoder appears to introduce overfitting. Could the authors clarify whether task representation learning provides an improvement rather than a trade-off between generalization and overfitting?

**Limitations:**

Yes.

**Strengths And Weaknesses:**

Strengths:

1. The idea of applying mutual information knowledge distillation in offline multi-agent reinforcement learning is interesting. The paper also provides solid theoretical analysis and supporting proofs for the proposed algorithm.

2. The experiments are comprehensive. The results persuasively validate the effectiveness of the proposed task representation learning framework.


Weaknesses:

1.  Utilizing LLMs to produce role labels appears to be less novel as a core contribution.. Even though the role labeling depends on the current large models (GPT-5), the main training procedure of role representation learning is similar to the task representation learning presented in Section 4.1.

2. While Fig.4 shows that incorporating role encodings improves the performance, it is still unclear whether the improvements are driven by the role representation learning or from the external knowledge injected by GPT-5 during label generation.

3. Relying on GPT-5 for labeling leads to practical issues, such as high cost and heavy prompting requirements, which limits its extensive application in real world settings.

4. The consistency and reliability of labels generated by LLMs are still concerning, which affect the robustness and reproducibility of the proposed method.

---

> ### Author Rebuttal · Authors · 2026-03-30
>
> Thank you for your insightful reviews. We hope our responses address your concerns. Additional experiments: https://www.notion.so/Additional-experimental-results-for-ICML-paper-4780-333c2fed721a803d9aa3fea9cbf2ccaf?pvs=21
>
> ### Q1 Novelty
>
> **D$^2$TR addresses the unique challenges of offline meta-RL in multi-agent settings, and we therefore believe its core contributions are novel.** Specifically:
>
> - **MIKD for decentralized task representation learning:** To our knowledge, D$^2$TR is the first to use mutual information knowledge distillation (MIKD) to transfer knowledge from an easier-to-learn centralized global task encoder to decentralized local task encoders, directly addressing decentralized task representation learning in MA settings.
> - **Combining role and task representations:** Prior meta-RL mainly learns task representations. Although role representations have been studied in MA settings, they are usually limited to single-task settings or multi-task settings with aligned objectives. D$^2$TR is the first to combine task and role representations in meta-RL with potentially conflicting reward functions across tasks. This overcomes the inability of task representations alone to capture heterogeneous agent roles within the same task, improving generalization to unseen tasks induced by role recomposition. The key contribution is **introducing role representations** into this setting, rather than the specific learning method.
> - **LLM-based role labeling:** Compared with the two contributions above, LLM-assisted role labeling is mainly **an enabling component** for the second contribution rather than a core contribution itself. Since the dataset lacks explicit role annotations, we use LLMs to infer them from semantic information. **D$^2$TR is not tied to this specific labeling method**: alternative methods can be readily incorporated.
>
> ### Q2 More analysis on LLM-based role annotation
>
> We added a sensitivity study on **LLM-based role annotation** in **Table 1 of the experiment link**. Our LLM-based role labeling is reasonably robust: performance is only mildly affected by **LLM selection**, **LLM randomness**, and **prompt wording**. Comparison with **clustering-based role labeling** without language shows that LLM prior knowledge is essential in some environments, while in others, **description-based clustering** can also facilitate knowledge sharing across tasks. Due to the character limit, more details are provided in our response to **Q2 of Reviewer r7As**.
>
> ### Q3 Cost for utilizing GPT-5
>
> We agree that computational cost and prompting effort are common challenges for LLM-based methods, and our role labeling method explicitly **balances the two**.
>
> Labeling each trajectory individually with the LLM would be prohibitively expensive. We therefore design a four-stage pipeline—pre-clustering, then description, summarization, and merging—which trades additional learning and prompt design for far fewer LLM calls. As shown in **Appendix C.2**, this reduces the number of LLM queries from **40,000** trajectory-level calls to about **300** in one example.
>
> Moreover, experiments on LLM choice show that a more cost-efficient model, e.g. **DeepSeek V3.2**, achieves comparable performance, suggesting that the overall cost can be further reduced.
>
> ### Q4 Typo
>
> We sincerely thank the reviewer for the careful review, we will revise Eq. 8, Eq. 9 accordingly.
>
> ### Q5 Utility of task representations
>
> We agree that if roles could be **perfectly inferred**, task representation might become redundant and could even hurt generalization, as also suggested by **Fig. 8 in Appendix E.2**. We nevertheless retain task representation for two reasons:
>
> - When **roles are hard to distinguish**, task representation may be the **only viable way** to identify tasks and support adaptation and generalization. Although our experiments mainly consider environments with clear roles, many real-world settings may lack an explicit notion of role; e.g., in robotic control, each agent may control the same joint across tasks while optimal strategies still differ, making roles hard to define meaningfully.
> - When **role inference is difficult**, task representation helps **balance exploration and exploitation**. This issue matters because role inference typically **depends on expert demonstrations**, which may be unavailable. In online adaptation, for example, no demonstrations are given, so the policy must infer the task from its own data. If only role representation is used, the role encoder must rely on initially random trajectories, making meaningful role recovery difficult. In contrast, the task encoder places weaker requirements on context quality because it mainly uses rewards and next-state signals. A better strategy is therefore to **use task representation for early exploration, then shift to role representation for exploitation once higher-quality trajectories are collected**, as also supported by **Fig. 8 in Appendix E.2**.

---

> > ### Author Rebuttal · Reviewer_xKZi · 2026-04-01
> >
> > Thank you for your response. I agree that your contributions of MIKD for decentralized task representation learning, along with the integration of role and task representations is novel. I also appreciate the additional experiments, particularly those evaluating clustering-based baselines without LLMs, helpful for understanding the method.
> >
> > That said, I still have a concern regarding the source of the observed performance gains. In particular, it remains unclear whether the improvements primarily arise from the role and task representation learning framework itself, or from the higher-quality labels provided by LLMs.
> >
> > Based on the additional results on the CN Target and Stalker–Zealot tasks, the clustering-based variants (0.24 vs. 0.28 on CN Target, and 0.67 vs. 0.65 on Stalker–Zealot) do not show substantial improvements over the backbone. In contrast, the LLM-based variants demonstrate more consistent gains (at least 0.37 on CN Target and at least 0.69 on Stalker–Zealot). These observations suggest that the external knowledge introduced by LLMs may play a more significant role than the representation learning mechanism alone.

---

> > > ### Author Response · Authors · 2026-04-02
> > >
> > > Thank you very much for your response and for your positive assessment of our rebuttal. We sincerely appreciate your feedback and hope that our response has addressed your additional questions.
> > >
> > > In fact, in our work, the introduction of **roles** and the use of **LLM-provided role labels** are complementary and both essential.
> > >
> > > From the perspective of role modeling, obtaining high-quality role representations first requires **reasonable and informative role labels**. However, in current meta-MARL settings, there is no ideal existing method that can reliably provide effective role labels in scenarios where task objectives may conflict and the state space may vary due to different numbers of agents. This is also why the **clustering-based role labels** do not perform as well in our experiments.
> > >
> > > In the **CN Target** environment, the role labels produced by clustering are largely difficult to interpret, meaning that they **provide little additional information**. As a result, introducing role representations based on these labels does not bring clear improvement. In the **Stalker-Zealot** environment, clustering-based roles can indeed **provide a finer-grained partition of agent races within the same task**. However, because the state spaces differ across tasks, the ability to share knowledge across tasks is still limited, which prevents clustering-based role labels from matching the performance of our proposed **LLM-based role labels**.
> > >
> > > Even so, as shown in the table below, **clustering-based roles** still improve OOD performance over both the **Backbone** baseline and the **w/o Role** ablation in the main paper, which confirms that the **introduced role information is indeed useful**. Overall, **the more informative the role labels are, the higher the quality of the resulting role representations, and thus the larger the overall performance gain**.
> > >
> > > | Task | Clustering Role | Backbone | w/o Role |
> > > | --- | --- | --- | --- |
> > > | 2s3z | **0.99 0.01** | 0.97 0.00 | **0.99 0.01** |
> > > | 2s4z | **1.00 0.00** | 0.97 0.00 | 0.97 0.02 |
> > > | 3s5z | 0.99 0.01 | **1.00 0.00** | 0.97 0.02 |
> > > | IID Mean | **0.99 0.01** | 0.98 0.00 | 0.98 0.02 |
> > > | 1s3z | **0.61 0.07** | 0.58 0.01 | 0.52 0.11 |
> > > | 1s4z | 0.83 0.09 | 0.80 0.11 | **0.91 0.02** |
> > > | 1s5z | 0.66 0.05 | **0.68 0.01** | 0.58 0.01 |
> > > | 2s5z | 0.83 0.07 | **0.96 0.01** | 0.93 0.04 |
> > > | 3s3z | **0.77 0.04** | 0.74 0.04 | **0.77 0.01** |
> > > | 3s4z | 0.91 0.00 | **0.94 0.00** | 0.91 0.00 |
> > > | 4s3z | **0.44 0.02** | 0.38 0.08 | 0.42 0.14 |
> > > | 4s4z | **0.50 0.07** | 0.42 0.07 | 0.31 0.02 |
> > > | 4s5z | **0.47 0.02** | 0.33 0.04 | 0.32 0.08 |
> > > | OOD Mean | **0.67 0.03** | 0.65 0.02 | 0.63 0.00 |
> > >
> > > Second, from the perspective of LLM prior knowledge, **introducing role serves as an important bridge between LLM prior knowledge and the meta policy**. Without role, and given prior knowledge alone, we are currently unaware of any other concise and efficient way to incorporate such knowledge into an offline meta policy so as to improve generalization performance; addressing this likely requires further investigation in future work. In contrast, our proposed role representation enables the use of LLM knowledge to improve policy performance in a cost-efficient manner, making it **one effective bridge for jointly leveraging knowledge and policy in a data-driven way**.
> > >
> > > In summary, in our work, **the introduction of role and the LLM-provided role labels are complementary and both indispensable**. While role is a core contribution, the proposed LLM role labeling is **equally important for fully realizing its benefits**. While we emphasize that our method is not tied to this specific labeling strategy, however, we believe LLM-provided role labels **offer a important and practical way to obtain high-quality role annotations**. If a future method can produce even **higher-quality and more cost-efficient** role labels, we would be very happy to combine it with our framework.

---

### Official Review · Reviewer_FKZb · 2026-03-13

**Soundness:** 3
**Presentation:** 3
**Significance:** 3
**Originality:** 3
**Overall Recommendation:** 4
**Confidence:** 4

**Summary:**

This paper presents a context-based meta-RL framework that enables efficient decentralized and disentangled identification of task and role information. It aligns decentralized task representations with centralized ones via mutual-information-based distillation and uses an LLM to assign roles in offline trajectories for improved decentralized role inference.

**Compliance With Llm Reviewing Policy:**

Affirmed.

**Key Questions For Authors:**

**Questions**

(Q1) I don’t think I fully understand the procedure of LLM assisted role labeling. Although text description is provided, presenting concise algorithm for this procedure may help readers. What are the $C_1, C_2, \cdot, C_r$? Are they clusters? How do we define $r$ properly? and is the proposed method sensitive to such $r$? How do we guarantee that?

In addition, what are the relation between $C_i$ and $PD$? Do we use a fixed $PD$ once we conduct pre-cluster?

(Q2) Related to Q1, do we have to use LLM iteratively to conduct such procedure? One could use various trajectory embeddings for clustering as in [1] if we already had fixed dataset.

[1] Na, Hyungho, et al. "Trajectory-class-aware multi-agent reinforcement learning." The Thirteenth International Conference on Learning Representations. 2025.

**Minor questions and comments**

(C1) Please mention corresponding proofs in the Appendix within the main manuscript for theorems and propositions.

(Q3) What is $j$ in $L_{M_i,j,e}$? Is it agent index? If so, why not use the same notation as in $\tau_i^j$?

(Q4) As the proposed method includes various components, it would be helpful to clarify how long it takes to train each component and the overall policy.

**Limitations:**

Limitations are not listed in the manuscript. Please see weaknesses.

**Strengths And Weaknesses:**

**Strength**

The paper is well motivated and in general well-written. The related work covers diverse fields and mostly they are sufficiently covered. The presented results seem meaningful, especially for OOD cases.


**Weakness**

The related work is well presented but as the paper mainly focuses on offline MARL settings, more recent works, such as HiSSD [1] whose code is available and BiKT [2] whose code is not available, should be included or at least discussed.

[1] Liu, Sicong, et al. "Learning generalizable skills from offline multi-task data for multi-agent cooperation." arXiv preprint arXiv:2503.21200 (2025).

[2] Zhang, Junkai, et al. "Bi-level knowledge transfer for multi-task multi-agent reinforcement learning." The Thirty-ninth Annual Conference on Neural Information Processing Systems. 2025.

---

> ### Author Rebuttal · Authors · 2026-03-30
>
> Thank you for your careful reviews and constructive suggestions. We appreciate your feedback and hope that our response addresses your concerns. Additional experiments are provided in [https://telling-floor-898.notion.site/Additional-experimental-results-for-ICML-paper-4780-333c2fed721a803d9aa3fea9cbf2ccaf](https://www.notion.so/Additional-experimental-results-for-ICML-paper-4780-333c2fed721a803d9aa3fea9cbf2ccaf?pvs=21)
>
> ### Q1 Comparison with HiSSD and discussion about BiKT
>
> Thank you very much for the valuable suggestion. HiSSD and BiKT are indeed important related works in the area of multi-agent skill discovery, and we will add corresponding citations and discussion in both the **Introduction** and **Related Work** sections. To enable a more thorough comparison with a stronger skill-based approach, we further include **HiSSD** as an additional baseline.
>
> The comparison with **HiSSD** is reported in **Table 2 in the experiment link**. The results show that **HiSSD** achieves better performance in the **Marine Hard** and **Terran** environments, whereas **D$^2$TR performs better in the remaining environments**. This suggests that both methods are strong and effective offline multi-task multi-agent approaches, with their relative advantages depending on the specific scenario.
>
> ### Q2 Questions about LLM assisted role labeling
>
> Providing explicit pseudocode for LLM-assisted role labeling would indeed improve the readability of the algorithm, and we sincerely appreciate this suggestion. Here, $C_1, C_2, \cdot, C_r$ denote the **final role labels output by the LLM**, i.e., the clustering result. The value of $r$ is **determined directly by the LLM rather than fixed in advance**; in the prompt, we only specify **an admissible range**, e.g., $4-9$.
>
> Take Stalker-Zealot as an example. This domain contains three tasks, namely $2s3z, 2s4z$ and $3s5z$. Pre-clustering divides each task into four clusters, so $PD$ is **fixed** as
>
> $PD=\\{2s3z_0,2s3z_1,\dots,2s4z_0,2s4z_1,\dots,3s5z_0,3s5z_1,3s5z_2,3s5z_3\\}.$
>
> After pre-clustering, we perform trajectory description and cluster description with the LLM for each cluster in $PD$. Based on the resulting cluster descriptions, the LLM finally merges them into the following clustering result:
>
> $0:\\{2s3z_0, 2s3z_1, 2s4z_2, 3s5z_0\\}, 1:\\{2s3z_3, 2s4z_1, 3s5z_2\\}, 2:\\{2s3z_2, 2s4z_0\\}, 3:\\{2s4z_3, 3s5z_1, 3s5z_3\\},$
>
> where $0,1,2,3$ correspond to $C_1,C_2,C_3,C_4$, respectively. That is, all trajectories belonging to $0$ cluster such as trajectories in $2s3z_0$ are assigned the same role label $0$.
>
> ### Q3 Should we use LLM iteratively
>
> Within each environment, the LLM is **invoked only once**, following a single pipeline of Pre-clustering$\to$LLM labeling.
>
> That said, an alternative approach would be to perform pre-clustering using multiple types of embeddings, apply LLM-based labeling to each resulting clustering, and then aggregate the labels via a voting scheme. While this strategy could potentially improve the robustness of role labeling, it would also incur substantially higher computational cost. We therefore leave this direction for future work.
>
> ### Q4 Mention proofs in the Appendix within the main manuscript
>
> Thank you very much for your valuable suggestion, we will mention proofs in the Appendix within the main manuscript to enable better readability.
>
> ### Q5 $j$ in $L_{M_i,j,e}$
>
> In fact, $j$ refers to the **clustering ID within** $PD$. For example, when
>
> $PD=\\{2s3z_0,2s3z_1,\dots,2s4z_0,2s4z_1,\dots,3s5z_0,3s5z_1,3s5z_2,3s5z_3\\},$
>
> the range of $j$ is $0,1,\dots,11$, since there are 12 clusters in total within $PD$. We sincerely apologize for the confusion caused by the overlap between this notation and the agent ID. We will revise the notation accordingly to avoid ambiguity.
>
> ### Q6 Training time analysis
>
> We sincerely thank the reviewer for the valuable suggestion. Using the Stalker-Zealot environment as an example, we report the per-task training time measured on a single NVIDIA 4090 GPU under an otherwise idle setting (i.e., no additional CPU/GPU workload), as summarized in the table below.
>
> | Task Encoder Training | Pre-clustering | Role Labeling | Role Encoder Training | Policy Training |
> | --- | --- | --- | --- | --- |
> | 1.67h/task | 1.09h/task | 0.45h/task | 1.17h/task | 6.31h/task |
>
> It can be observed that **the majority of the computational cost is still dominated by the policy training stage** (which also includes evaluation on both IID and OOD tasks). In comparison, the combined runtime of role labeling and role encoder training is on par with that of task encoder training. Meanwhile, pre-clustering does not introduce significant overhead, as it can be parallelized across tasks.

---

> > ### Author Rebuttal · Reviewer_FKZb · 2026-04-03
> >
> > I sincerely appreciate the authors' efforts in preparing this rebuttal. My questions have been properly answered, and several of my concerns have been resolved. The authors also provided additional experiments, which seem to ensure a fair comparison. Although the revised manuscript includes several improvements that are not yet available for review, I believe the authors will address them properly in the final version.

---

> > > ### Author Response · Authors · 2026-04-03
> > >
> > > We would like to express our sincere gratitude for your time and effort dedicated to reviewing our submission. We feel glad to address your concerns and grateful for your recognition.

---

### Official Review · Reviewer_r7As · 2026-03-23

**Soundness:** 3
**Presentation:** 3
**Significance:** 3
**Originality:** 3
**Overall Recommendation:** 4
**Confidence:** 4

**Summary:**

This paper studies **generalizable offline multi-agent meta-reinforcement learning under decentralized execution**. To this end, it proposes the **D2TR** framework, which aims to learn disentangled task and role representations from offline multi-task datasets. For task identification, the method extracts a centralized task representation from global trajectories and distills it into a decentralized task encoder via a mutual information (MI)-based knowledge distillation objective. For role modeling, the paper leverages a large language model (LLM) to assign role labels to trajectories, and then trains a role encoder through contrastive learning. The resulting task and role representations are further fed into a Transformer-based offline multi-agent meta-RL policy learner, which is built upon an OMIGA-style actor-critic objective. Experiments on the CN, SMAC, and SMACv2 benchmarks show that, compared with several multi-task and offline meta-RL baselines, the proposed method effectively improves out-of-distribution (OOD) generalization.

**Compliance With Llm Reviewing Policy:**

Affirmed.

**Final Justification:**

Thank you for the detailed response which addresses my concerns. I am therefore maintaining my positive score.

**Key Questions For Authors:**

1. **How robust is the LLM-based role annotation pipeline?**
    Please clarify how sensitive this process is to prompt wording, random seeds or sampling strategies, and the choice of different LLMs.
2. **Can the authors provide a role annotation baseline that does not rely on LLMs?**
    For example, clustering-based labels, simple heuristic labels, or other automatically generated labels would be useful baselines. This is important for demonstrating that the performance gains come from semantic role discovery rather than simply from introducing additional supervision.
3. **Can the authors provide a more rigorous justification for Theorem 4.1?**
    In particular, the independence assumptions in Appendix B.1 would benefit from a stronger and more careful argument.

**Limitations:**

1. The dependence on external LLMs for label generation and the resulting reproducibility concerns;
2. The additional computational cost and implicit supervision introduced by LLM-assisted annotation;
3. The currently limited evidence for transfer beyond benchmark game environments.

**Strengths And Weaknesses:**

## Strengths
1. **The problem setting is clearly defined and practically meaningful.**
    The paper focuses on offline multi-agent meta-RL with decentralized execution and OOD generalization, which is an important and worthwhile research direction with clear practical relevance.
2. **The experimental evaluation is fairly comprehensive.**
    The paper reports results on multiple benchmark datasets, including CN, SMAC, and SMACv2, covering both IID and OOD settings. It also includes online adaptation curves, several ablation studies, and rich visualization results. Overall, the experimental effort appears substantial.
3. **The method shows clear gains in OOD settings.**
    Compared with the baselines, the proposed approach achieves noticeable improvements in OOD scenarios. The role ablation results also suggest that role modeling can help improve generalization in certain environments.

## Weaknesses
1. **The theoretical derivation relies on overly idealized assumptions.**
    The key theoretical results depend on strong approximations, such as the conditional independence assumption in Theorem 4.1, whose justification appears largely heuristic. As a result, the mathematical rigor of the analysis is not fully convincing.
2. **The LLM-based role annotation pipeline is insufficiently validated.**
    Role labels are a core component of the method, yet the paper does not provide sufficient analysis of their robustness with respect to prompt design, LLM choice, randomness, or annotation noise. It remains unclear whether this pipeline is stable and reproducible, or whether the observed gains depend heavily on prompt engineering. In addition, the paper lacks comparisons with simpler alternatives, such as clustering-based labels, heuristic labels, or other role-discovery baselines.
3. **The absolute performance in complex environments and the quantitative validation of “disentanglement” remain limited.**
    In some more challenging environments, the absolute improvement in generalization is still relatively modest. Moreover, the central claim of disentanglement is currently supported mainly by visualization, without more direct quantitative evidence—for example, whether the task representation is invariant to role information, and whether the role representation is invariant to task information. Finally, the overall pipeline appears somewhat complicated and cumbersome.

---

> ### Author Rebuttal · Authors · 2026-03-30
>
> Thank you for your thoughtful reviews. We hope that our response addresses your concerns. Additional experiments are provided in [https://telling-floor-898.notion.site/Additional-experimental-results-for-ICML-paper-4780-333c2fed721a803d9aa3fea9cbf2ccaf](https://www.notion.so/Additional-experimental-results-for-ICML-paper-4780-333c2fed721a803d9aa3fea9cbf2ccaf?pvs=21)
>
> ### Q1 Strong assumption in Theorem 4.1
>
> We thank the reviewer for pointing this out. Appendix B.1 originally relied on an overly heuristic conditional-independence argument. In the revision, we will **replace it with an exact information-theoretic decomposition**:
>
> $I(z_i^j; M, z_M) = I(z_i^j; z_M) + I(z_i^j; M | z_M)= I(z_i^j; M) + I(z_i^j; z_M | M),$
>
> which implies
>
> $I(z_i^j; z_M) - I(z_i^j; z_M | M) \le I(z_i^j; M) \le I(z_i^j; z_M) + H(M | z_M),$
>
> that is
>
> $|I(z_i^j; M)-I(z_i^j; z_M)|\le I(z_i^j; z_M | M)+H(M | z_M).$
>
> Thus, the distillation term $I(z_i^j;z_M)$ is a **gap-controlled surrogate** of the target objective $I(z_i^j;M)$, and the upper-gap term is reduced by maximizing $I(z_M;M)$ since $H(M | z_M) = H(M) - I(z_M; M)$. Importantly, our practical local objective in Eq. (9) still directly optimizes an approximation of $I(z_i^j;M)$. Theorem 4.1 is meant to justify the alignment of the additional distillation term, rather than a strict replacement of the local objective. Deriving a tighter bound independent of $I(z_i^j; z_M | M)$ remains promising future work.
>
> ### Q2 More analysis on LLM-based role annotation
>
> Thank you for the valuable suggestion. We have added a sensitivity study on **LLM-based role annotation** in **Table 1 of the experiment link**.
>
> First, comparing our method with variants using **DeepSeek V3.2** and **GPT-5-mini** shows that, in both the **code-based role** setting on **CN Target** and the **description-clustering-based** setting on **Stalker-Zealot**, the choice of LLM has only a limited effect on overall performance. This suggests that our method is reasonably robust to the specific LLM used.
>
> Second, comparison with a **clustering-based baseline without LLMs** reveals clear environment-dependent differences. In **CN Target**, **Clustering Role** generalizes poorly and does not outperform the **Backbone**, highlighting the need for external knowledge in role labeling. In **Stalker-Zealot**, **Clustering Role** is slightly worse than our method but still better than the Backbone, showing that role information is beneficial. However, clustering can only separate agents of different races within a task and cannot effectively merge same-race agents across tasks, limiting knowledge sharing and likely explaining its gap from our **LLM-based description** method.
>
> Finally, in **description-clustering-based Stalker-Zealot**, we add two more baselines: **New Seed**, which asks the LLM to perform a fresh round of role labeling, and **Simple Prompt**, which removes the example (*“for example, a tank that lures enemies on the front line, or a kiter dealing damage from the back line”*) in our prompt. Both achieve performance similar to the full method, suggesting that our approach is also reasonably robust to **LLM randomness** and **prompt wording**.
>
> ### Q3 Quantitative validation of disentanglement
>
> While task and role are correlated, the correlation is **insufficient for one to fully represent the other**. Take **CN Target** as an example: it has 20 uniformly distributed tasks and 6 uniformly distributed roles, and each task is determined by 3 roles. Thus, $H(\text{role})=\log_2 6\approx 2.585$ bits and $H(\text{task})=\log_2 20\approx4.322$ bits. Given a task, the role still has 3 possible values, so $H(\text{role}\mid\text{task})=\log_2 3$. Therefore, the mutual information is $I(\text{role};\text{task})=\log_2 2=1$ bit, meaning that observing the task reduces role uncertainty by only $\frac{I(\text{role};\text{task})}{H(\text{role})}\approx 0.387$. Hence, task information alone cannot fully characterize role, which motivates introducing a dedicated role representation to disentangle task and role. At the same time, task still contains partial role information, justifying our use of task representation as an input to the role encoder.
>
> ### Q4 Complicate pipeline
>
> Learning task representations is common in meta-RL. Our pipeline can therefore be viewed as **augmenting standard meta-RL with role labeling and role representation learning**, and the gains from role representations justify this added complexity. That said, **developing simpler and more efficient methods for role labeling and representation learning remains a promising direction**, with potential for further improvement.
>
> Moreover, the **time-complexity results in Q6 to Reviewer FKZb** show that the combined runtime of **role labeling and role encoder training is comparable to that of task encoder training**. The main extra overhead comes from pre-clustering, but since it can be parallelized across tasks, the **overall additional cost remains modest**.

---

> > ### Author Rebuttal · Reviewer_r7As · 2026-04-04
> >
> > Thank you for the detailed response which addresses my concerns. I am therefore maintaining my positive score.

---

> > > ### Author Response · Authors · 2026-04-04
> > >
> > > We would like to express our sincere gratitude for your time and effort dedicated to reviewing our submission. We feel glad to address your concerns and grateful for your recognition.

---

### Decision · Program_Chairs · 2026-04-30

**Decision:**

Accept (regular)

**Comment:**

This paper proposes Decentralized and Disentangled Task–Role Representation Learning (D^2TR), a framework for offline multi-agent meta-RL that learns disentangled task and role representations to support decentralized execution and out of distribution generalization. Core contributions include mutual information-based knowledge distillation (MIKD) to align centralized and decentralized task encoders, and LLM-assisted role labeling to supervise contrastive role representation learning.

The problem is well-motivated, the MIKD approach is novel, and the experimental evaluation is comprehensive with convincing out of distribution gains.

Nevertheless, there are concerns about about the presentation, some of the assumptions, and the significance of the work. For instance,
it is difficult to determine whether performance gains are from the representation learning framework or from the external knowledge injected by LLM-generated labels, as clustering-based alternatives underperform substantially. Furthermore, Theorem 4.1 relies on assumptions that are motivating rather than rigorous. And the disentanglement claim is supported visually but not quantitatively. The LLM dependency also raises reproducibility and applicability concerns, though the added sensitivity study is reassuring. Also, note that policies for Dec-POMDPs can use single observations but should be defined in terms of histories (which often allow better performance).

The rebuttal was thorough and resolved most concerns but any future version of the paper should incorporate the promised revisions, including the updated theoretical treatment, sensitivity experiments, and other presentation fixes.